# Context coding in the mouse nucleus accumbens modulates motivationally relevant information

**Jimmie M. Gmaz** [ORCID], **Matthijs A. A. van der Meer** [ORCID] *

Department of Psychological and Brain Sciences, Dartmouth College, Hanover, United States of America

* mvdm@dartmouth.edu

## Abstract

Neural activity in the nucleus accumbens (NAc) is thought to track fundamentally value-centric quantities linked to reward and effort. However, the NAc also contributes to flexible behavior in ways that are difficult to explain based on value signals alone, raising the question of if and how nonvalue signals are encoded in NAc. We recorded NAc neural ensembles while head-fixed mice performed an odor-based biconditional discrimination task where an initial discrete cue modulated the behavioral significance of a subsequently presented reward-predictive cue. We extracted single-unit and population-level correlates related to the cues and found value-independent coding for the initial, context-setting cue. This context signal occupied a population-level coding space orthogonal to outcome-related representations and was predictive of subsequent behaviorally relevant responses to the reward-predictive cues. Together, these findings support a gating model for how the NAc contributes to behavioral flexibility and provide a novel population-level perspective from which to view NAc computations.

## Introduction

The nucleus accumbens (NAc) is an important contributor to the motivational control of behavior, acting directly through output pathways involving brainstem motor nuclei ("limbic-motor interface") [1–3] and indirectly through return projections within cortico-striatal loops [4,5]. Accordingly, leading theories of NAc function, and the mesolimbic dopamine (DA) system it is tightly interconnected with, tend to focus on the processing of reward (and punishment) and its dual role in energizing and directing ongoing actions as well as in learning from feedback [1,6–9]. These proposals attribute to the NAc a role in motivational and reward-related quantities such as incentive salience, value of work, expected future reward, economic value, risk and reward prediction error. In more formal reinforcement learning models, the NAc-dopamine system is typically cast as an "evaluator" or "critic," tracking state values that are useful to set the value of work as well as a source of a teaching signal in the form of reward prediction errors [10–14]. Although the specifics are the subject of vigorous debate, these prominent theories all share a fundamentally **value-centric** focus: Notwithstanding substantial

**Data Availability Statement:** All preprocessed data files are available on GIN: https://gin.g-node.org/jgmaz/BiconditionalOdor.

**Funding:** This work was supported by Dartmouth College (start-up funds to M.A.A.v.d.M.) and the

Natural Sciences and Engineering Research Council (NSERC) of Canada (Canada Graduate Scholarship to J.M.G.). The funders had no role in study design, data collection and analysis, decision to publish, or preparation of the manuscript.

**Competing interests:** The authors have declared that no competing interests exist.

**Abbreviations:** BOLD, blood-oxygen-level-dependent; dPCA, demixed principal component analysis; LDA, linear discriminant analysis; mPFC, medial prefrontal cortex; NAc, nucleus accumbens; OFC, orbitofrontal cortex; PCA, principal component analysis; PETH, peri-event time histogram.

heterogeneity in NAc cell types and circuitry [6,15,16], this brain structure as a whole is typically cast as tracking a relatively low-dimensional quantity: a value signal that at its simplest is just a single number, reflecting how good or bad the current situation is.

These value-centric accounts are supported by a vast literature demonstrating that NAc manipulations can exert bidirectional control over motivated behaviors such as conditioned responding to reward-predictive cues and regulate how much effort to exert [7,17–21] as well as the observation that electrical or optogenetic stimulation of the NAc itself, or dopaminergic terminals in the NAc, is sufficient for inducing behavioral preferences [16,22–27]. Similarly, unit recording studies in rodents and fMRI work in humans consistently report widespread, sizable value signals in NAc single units, populations, and the NAc blood-oxygen-level-dependent (BOLD) signal [28–38]. Thus, there seems to be widespread agreement that the major dimension (principal component) of NAc activity is some form of value signal.

However, in complex, dynamic behavioral tasks, lesions or inactivations of the NAc lead to deficits that are not straightforward to explain from a purely value-centric perspective [6,39], such as the implementation of conditional rules [40], or switching to a novel behavioral strategy [41]. In addition, prominent inputs from brain regions such as orbitofrontal cortex (OFC), medial prefrontal cortex (mPFC), and hippocampus [42–44] suggest that the NAc has access to nonvalue signals that would be expected to not only inform its function but help shape its neural activity. Indeed, a study in primates suggests that elements of task structure, which are orthogonal to value, but nonetheless crucial for successful behavior on the task, are represented in NAc [45]. In rodents, there have been hints of task structure too, but this has been hard to show conclusively due to the difficulty in cleanly dissociating task structure from value [46,47] (see also related work on dopamine neuron and OFC activity representing task structure [42,48]). The distinct states that make up the structure of these and other tasks are often referred to as "rules" or "contexts" that are learned from experience and require the inference and/or maintenance of information not currently presented as a sensory cue. Thus, it is currently unknown if, and how, task structure is encoded in rodent NAc, and if found, how such a signal relates to the subsequent processing of motivationally relevant information.

To address this issue, we trained mice to perform a biconditional discrimination task, where we model context as a task state signaled by one of 2 discrete odor cues. Specifically, animals were presented with 2 different "context" cues that determined whether a subsequent "target" cue would be rewarded (Fig 1A). Thus, in context O1, O3 but not O4 is rewarded, whereas in context O2, O4 but not O3 is rewarded. We recorded ensembles of NAc neurons and tested whether there is coding of the (equally valenced) context cues at the single cell and population level. Next, we used contemporary population analysis tools to test if this context signal can be used to inform subsequent behaviorally relevant processing of target cues.

## Results

### Mice learn to perform a biconditional discrimination task using odor cues

We sought to test whether NAc encodes information about task structure that is independent of reward. To do this, we used a biconditional discrimination task in which the identity of a "context" cue determines whether a subsequent "target" cue is rewarded or not [49–51]. We use the term "context" here to mean a cue that modifies the meaning of a subsequently presented target cue (i.e., whether that target cue predicts reward or not; see Discussion). Briefly, a trial began with presentation of one of the 2 context cues for 1 s, followed by a 2-s delay, followed by presentation of one of the 2 target cues for 1 s, followed by an additional 1-s response period (Fig 1). Animals had to make a licking response either during presentation of the target cue or the subsequent response period to get a sucrose reward for rewarded cue pairings. For

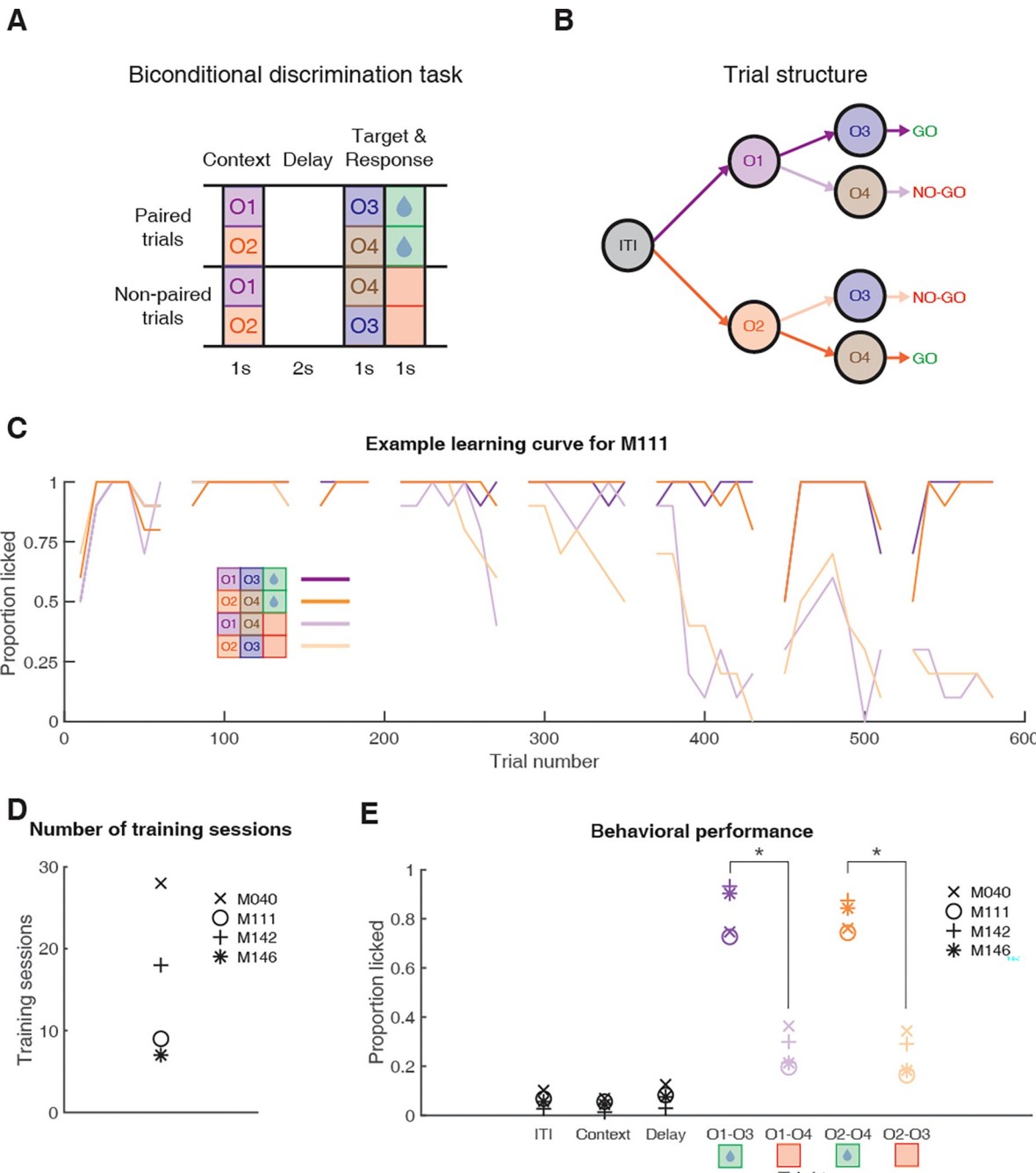

**Fig 1. Schematic of the behavioral task and results.** (**A**) Mice were trained in a head-fixed biconditional discrimination task where they learned to discriminate between different pairings of "context" and "target" cues. Mice were first presented with the context cue (1 s), followed by a delay (2 s), followed by target cue presentation (1 s), and an additional response period (1 s). Whether a target cue was rewarded depended upon the identity of the preceding context cue. For instance, licking in response to O3 was rewarded when preceded by O1, but not O2, while for O4 was rewarded when preceded by O2, but not O1. Note, this means that by design, each context cue was rewarded on half of the trials it was presented on. (**B**) Trial structure of the task. Purple arrows indicate trials with context cue 1 (O1); orange arrows indicate trials with context cue 2 (O2). Dark arrows after context cue presentation indicate rewarded trials; light arrows indicate unrewarded trials. This color scheme is used throughout the text. (**C**) Example learning curve showing proportion of trials with a lick over the course of full-task training. Data are shown for each trial type, in 20 trial blocks, with gaps separating individual training sessions. This learning curve shows that by day 6 (around trial 400), there was a clear difference in responding to rewarded versus unrewarded trials. (**D**) Number of training sessions before each mouse reached the criterion of 3 consecutive sessions with >80% correct responding, after which recordings began. (**E**) Behavioral performance during recording sessions showing the average proportion of trials with a licking response during the ITI, context cue period, delay period, and target cue period for each trial type for each mouse,

demonstrating that mice discriminated between rewarded (green) and unrewarded trial types (red). Asterisks denote significant differences ($p < 0.05$ based on a bootstrap with trial labels shuffled; see Methods). Data: https://gin.g-node.org/jgmaz/BiconditionalOdor. ITI, intertrial interval.

example, given context cue O1, target cue O3 but not O4 is rewarded, but following context cue O2, O4 but not O3 is rewarded. Thus, by design, rewarded trial types O1 to O3 and O2 to O4 both had the same outcome value (future expected reward), while unrewarded trial types O2 to O3 and O1 to O4 also had the same outcome value, with the specific odor associations counterbalanced across mice. Importantly, a 2-s delay separated the 2 odor cues in a trial such that mice had to maintain a representation of the context cue while waiting for the target cue.

Mice ($n = 4$) completed a total of 7 to 28 training sessions to reach criterion before recording sessions began (see Fig 1C for an example learning curve; Fig 1D for number of training sessions for each mouse). During recording sessions, mice licked for a significantly larger proportion of rewarded trials than unrewarded trials (Fig 1E; proportion of rewarded trials with a lick response: 0.82 +/− 0.08 SD; proportion of unrewarded trials with a lick response: 0.26 +/− 0.08 SD; z-score across mice and sessions: 11.05; $p < 0.001$), but licked similarly across context cues for both rewarded trial types (O1 to O3: 0.83 +/− 0.11 SD; O2 to O4: 0.81 +/− 0.06 SD) and unrewarded trial types (O1 to O4: 0.27 +/− 0.08 SD; O2 to O3: 0.25 +/− 0.09 SD; z-score across mice and sessions: 0.51; $p = 0.61$). Furthermore, individual mice showed a similar level of correct responding to the target cues during recording sessions (M040: 70% +/− 9% SD; M111: 78% +/− 4% SD; M142: 80% +/− 5% SD; M146: 83% +/− 8% SD), and minimal licking to the context cues themselves (Fig 1E, S1 Fig; proportion of trials with a lick response during context cue presentation: M040: 10% +/− 12% SD; M111: 7% +/− 4% SD; M142: 3% +/− 1% SD; M146: 6% +/− 2% SD). Therefore, mice learned the appropriate context-target cue associations in the task.

## NAc single units signal context

We set out to determine if context is encoded by the NAc, particularly whether the NAc can discriminate between separate context cues that are equal in future predicted reward, and whether this discrimination persists during the delay period after cue offset. We recorded a total of 320 units with >200 spikes (out of 386 total units) in the NAc from 4 mice over 41 sessions (range: 8 to 12 sessions per mouse) during performance on the task. Initial inspection of the data revealed a diversity of single-unit responses, including units that showed transient responses to all odor cues regardless of their significance in the task, units that discriminated the various cue identities and their associations, and of particular relevance, units that showed a sustained discrimination between the context cues during the delay period (see Fig 2 for examples). The main cue features of interest in this task were context cue coding, context coding during the delay period after context cue offset, target cue coding, and rewarded versus unrewarded trial coding after target cue onset (behavioral relevance). To determine whether a given single unit encoded each feature, we focused on comparing the trial-averaged firing rate differences during periods surrounding context and target cue presentations (Fig 3). To determine general coding for odor cues, we compared the 1 s preceding and following cue presentation for all trial types. To investigate coding of cue features, we compared the 1 s of cue presentation for context and target cue coding, and the 1 s preceding target cue presentation for context delay coding. Out of all units included in the analysis, 80 (Fig 3B; 26%) discriminated between the 2 context cues when analyzed during cue presentation, and 49 (Fig 3C; 15%) discriminated between the 2 contexts when analyzed during the following delay period, showing that individual units within the NAc code for context. Importantly, as there were no

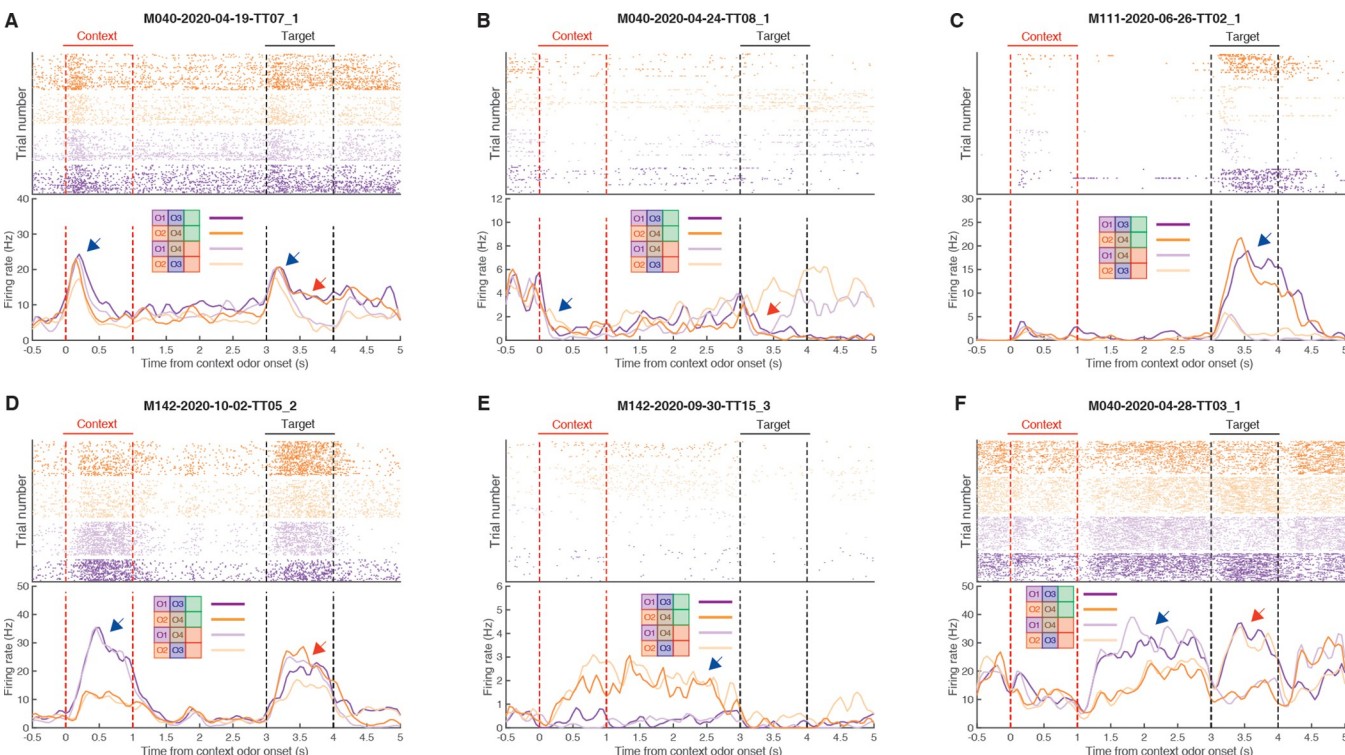

**Fig 2. Example single-unit responses for units with outcome- (top) and context-related (bottom) correlates.** Top of each plot shows spike rasters for the 4 trial types (purple: trials with context cue O1; orange: trials with context cue O2; dark colors: rewarded trials; light colors: unrewarded trials). Bottom half of each panel shows trial-averaged firing rates for each trial type aligned to context cue onset. Context cue presentation (0–1 s) is bordered by red lines, and target cue presentation (3–4 s) is bordered by black lines. (**A**) Example unit that shows a general response to cue presentation (blue arrow), as well as a subsequent discrimination between rewarded and unrewarded trials after target cue onset (red arrow). (**B**) Example unit showing a dip in firing after context cue onset (blue arrow), followed by a ramping of activity leading up to target cue onset, and a subsequent dip in firing after presentation of the rewarded target cue (red arrow). (**C**) Example unit that predominantly responds during presentation of the rewarded target cue (blue arrow). (**D**) Example unit that shows transient responses to the cues, showing a discrimination to both context (blue arrow) and target (red arrow) cues. (**E**) Example unit that discriminates between context cues, including throughout the delay period (blue arrow). (**F**) Example unit that discriminates context cues only during the delay period following offset of the context cue (blue arrow), as well as discriminating the subsequent target cues (red arrow). Data: https://gin.g-node.org/jgmaz/BiconditionalOdor.

differences in behavioral performance for the 2 context cues (Fig 1; z-score across mice and sessions: 0.51; $p$ = 0.61), this context coding can not be explained by differences in the perceived value of the cues. Apart from context coding, 216 units (Fig 3A; 68%; 153 increasing, 63 decreasing) showed a change in firing activity in response to both context cues relative to a precue baseline, 199 units (Fig 3D; 62%; 149 increasing, 50 decreasing) showed a change in firing activity in response to both target cues relative to a precue baseline, 87 units (Fig 3E; 27%) discriminated between the 2 target cues, and 97 units (Fig 3F; 30%; 77 increasing, 20 decreasing) discriminated between rewarded and unrewarded trials during target cue presentation. Finally, to assess changes in firing rate throughout a trial, we correlated the trial-averaged firing rates across all time points and found a general correlation between activity during context and target cue presentation, reflecting the large proportion of units that respond nondiscriminately to any cue (Fig 3H). Together, these results suggest that NAc encodes the various motivationally relevant features of the task, including NAc units that discriminated between the context cues during the delay period, suggesting that the NAc maintains information about which context the animal is in after offset of the cue itself.

Across mice, there appeared to be a qualitative relationship between the time spent in training and the strength of context cue coding, with mice that spent more time to acquire the task

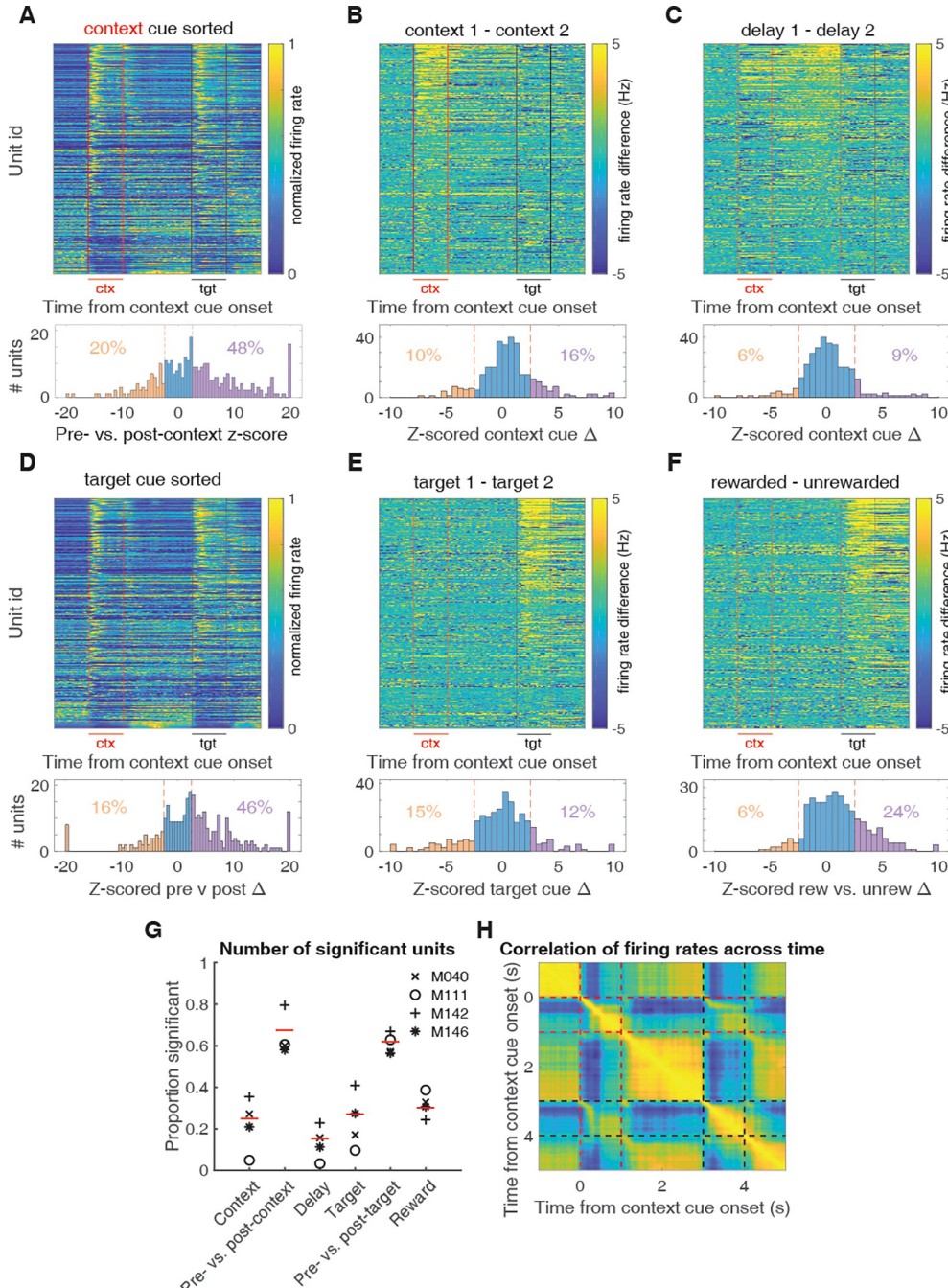

**Fig 3. Characterization of single-unit responses to the task.** Top of each plot is a heat plot showing either max normalized firing rates or firing rate differences for trial-averaged data for all eligible units, with unit identity sorted according to the peak value for the comparison of interest. Red lines border context cue presentation, and black lines border target cue presentation. Bottom of each plot shows a distribution of units with significant tuning for each task parameter, relative to a shuffled distribution. Red dotted lines signify z-scores of +/− 2.58. (**A**) Firing rate profiles for units at 1 s pre- and postcontext cue onset, sorted according to maximum value after context cue onset. (**B**) Firing rate differences for units across context cues, sorted according to maximum difference during context cue presentation. (**C**) Firing rate differences for units across context cues during the delay period, sorted according to maximum difference during the 1-s period preceding target cue presentation. (**D**) Firing rate profiles for units at 1 s pre- and posttarget cue onset, sorted according to maximum value after target cue onset. (**E**) Firing rate differences for units across target cues, sorted according to maximum difference during target cue presentation. (**F**) Firing rate differences for units for rewarded and unrewarded trial types during target cue presentation, sorted according to maximum difference during target cue presentation. (**G**) Proportion of significant units for each task component. Each mouse is indicated by a

symbol, and the average across mice is indicated by the red line. Note the higher context coding for M040 and M142, relative to M111 and M146. Also, note the stronger value coding in M111. General cue, gating, and state value categories represent units that were modulated by a combination of different task components; see Methods for classification details. (**H**) Correlation of firing rates across time on trial-averaged data across all units. Note that high correlations for periods of time when a cue is present, and the anticorrelations of these cue periods with precue periods. Red lines border context cue presentation, and black lines border target cue presentation. Data: https://gin.g-node.org/jgmaz/BiconditionalOdor.

showing stronger context coding than mice that had a shorter learning curve (S2 Fig). For instance, M040 (28 training sessions; 27% context coding units) and M142 (18 training sessions; 35% context coding units) showing more context-sensitive units than M111 (9 training sessions; 5% context coding units) and M146 (7 training sessions; 21% context coding units). Additionally, M111, which had the least amount of context coding units, also had the most caudal recording coordinates across mice (see Methods), but the strength of context coding did not follow a rostral-caudal gradient for the other mice. Furthermore, this variability across animals was not related to behavioral performance during recording sessions, and a similar relationship was also not seen for target cue-related coding. Together, this suggests that variability in context coding across mice might be due to differences in training duration or precise recording location.

A possible functional role for this context signal is to appropriately gate the response to cues whose relevance is context dependent. In this case, context-dependent activity preceding target cue presentation should be able to predict the behavioral response for a given target cue. For example, if a unit shows a higher firing rate for context cue O1 over O2, this discrimination would be linked to subsequent behavior if on a trial-by-trial basis it informed whether or not an animal licked in response to O3. In this situation, a licking response to O3 would be predicted on trials where the unit had a higher firing rate preceding target cue presentation. To test for this at the single-unit level, we first reran our firing rate comparisons across the whole trial period and found that 15% to 26% of units had firing rates that discriminated between the 2 context cues during the span of the context cue and delay periods (Fig 4A). We then trained a binomial regression to predict the behavioral response (lick or no lick) for a given target cue using the firing rate of a unit at various time points in a trial. We found that 5% to 13% of units (including 65% and 78% of all context- and delay-coding units, respectively) across time points were able to predict subsequent response to a target cue above chance, suggesting that individual units possess some information about the context that informs future response behavior (Fig 4B).

While single-unit analyses are informative to get a sense of what information is present within a neural population, the utility of these responses are dependent upon their position within the broader NAc network, and how they are interpreted by downstream structures. To characterize this population-level activity, we combined across-session data for each mouse to generate 4 pseudo-ensembles, one for each mouse in the dataset. As a first step to test if the population could improve predictions of trial outcome above and beyond that of the best performing single-unit units, we trained a binomial regression to predict the behavioral response for each target cue using the firing rates of these pseudo-ensembles (Fig 4C). This analysis revealed the ability to accurately predict the subsequent behavioral response for a given target cue above chance (prediction accuracy at most significant time point, M040: 82%; z-score: 5.82; $p < 0.001$; M111: 69%; z-score: 2.40; $p = 0.016$; M142: 80%; z-score 5.26; $p < 0.001$; M146: 75%; z-score: 4.13; $p < 0.001$). The variability observed across mice closely followed the observations from the single-unit data, with the M111 pseudo-ensemble data performing the worst at predicting the subsequent behavioral response. Thus, NAc population activity during

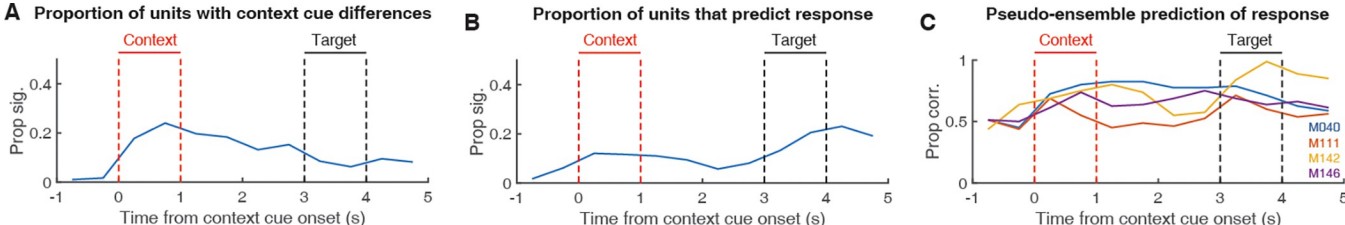

**Fig 4. Predicting behavioral response to a given target cue based on firing rate activity.** For a given target cue such as O3, this analysis sought to predict whether a lick or no lick response occurred based on activity preceding the target cue during the context and delay period. (**A**) Sliding window demonstrating the proportion of units that show discriminatory activity between the 2 context cues throughout the different periods in the trial, similar to Fig 3B and 3C. Red lines border context cue presentation, and black lines border target cue presentation. (**B**) Proportion of all units whose firing rate at a given point in the trial predicts the behavioral response to a given target cue above chance, according to a binomial regression. (**C**) Same as B, but using the firing rates of all units recorded for a mouse to generate a pseudo-ensemble prediction of the behavioral response for a given target cue. Each line denotes the prediction for a given mouse. Note, the variability across mice reflects the variability in single-unit context coding seen in Fig 3G. Data: https://gin.g-node.org/jgmaz/BiconditionalOdor.

the context and delay periods contain information relevant for the animal's subsequent context-dependent behavior.

## Multiple signals coexist within the context and delay period

The above analysis is a useful starting point in demonstrating that the behavioral response to a given target cue can be predicted from NAc activity, but it cannot determine *which components* of NAc activity underlie this prediction. One possibility is that the context signal specifically predicts the subsequent response; this notion would be one realization of the idea that NAc functions as a "switchboard" in which different context-specific activity patterns can gate the response to a given input [52–54] (Fig 5A). Alternatively, NAc activity is known to reflect expected future reward, referred to as "state value" in reinforcement learning, useful for setting the value of work and for the calculation of reward prediction errors [12,55,56]. Such ramp neurons that increase their firing rate as an animal approaches a reward site may interpret the context cues in our task as a state closer to reward, regardless of its identity (Fig 5B). Finally, a third possibility is that the NAc may generalize across all motivationally relevant cues (Fig 5C); any of these components could in principle drive the behavioral and neural response to the target cue.

To determine the relative strength of these hypothetical coding scenarios (Fig 5) in NAc activity and to test which component(s) drive the behavioral predictions (Fig 4C), we used the dimensionality reduction technique demixed principal component analysis (dPCA) to extract the task-related latent factors relating to the context cues, target cues, and their interaction (Fig 6). dPCA was the method selected as it constrains dimensionality reduction to extract the components that explain the most variance in the data for a given task parameter. dPCA differs from principal component analysis (PCA) as the latter extracts the components that capture the most variance in the data, agnostic to any aspects of the task. Additionally, dPCA was chosen over linear discriminant analysis (LDA), as LDA is focused on reconstructing identities, while dPCA is focused on reconstructing data means, and, thus, dPCA is better suited to preserve aspects of the original data. We applied dPCA to the pseudo-ensemble data from each mouse individually and extracted the top components related to each major task component (see Fig 7 for relative contributions of each component). This analysis revealed a variety of time-varying components, capturing different aspects of the task (see S3 Fig for the top 10 components from each animal). To test components that differ across trial types, we compared the components extracted from the data to components from a shuffled distribution extracted

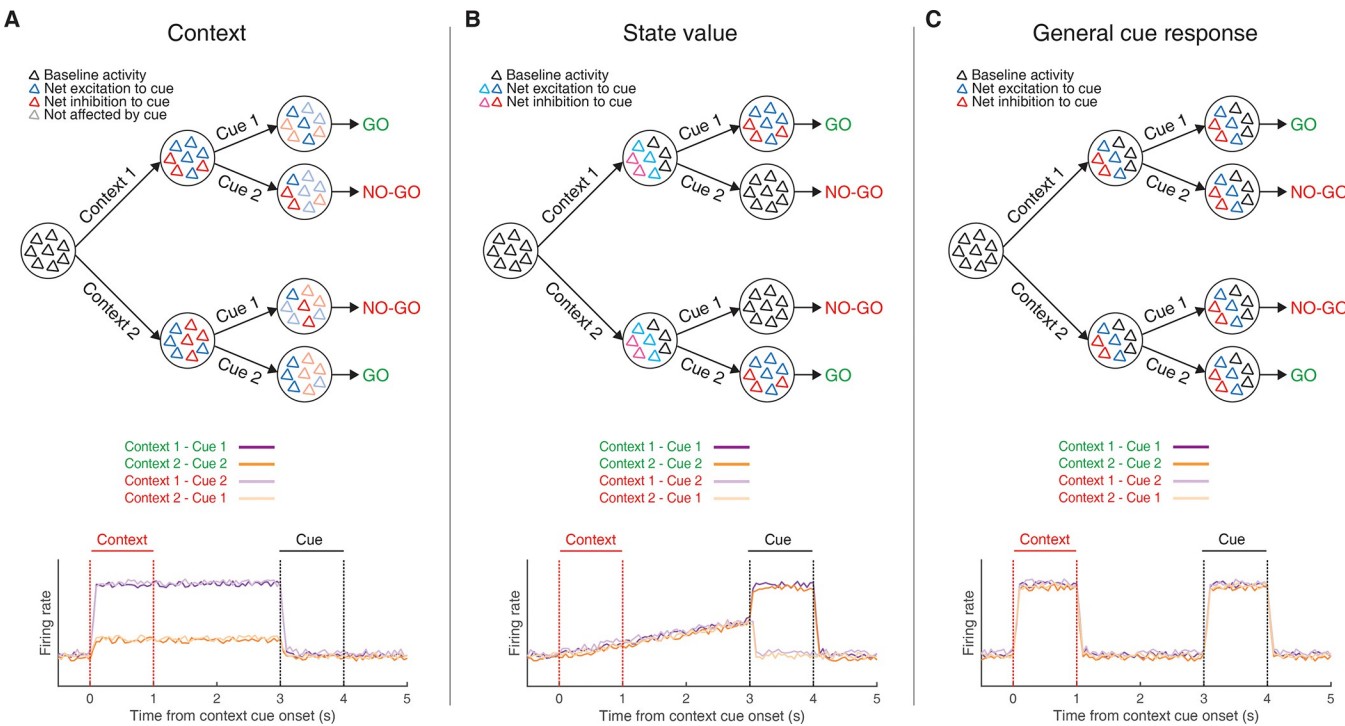

**Fig 5. Schematic of hypothetical coding scenarios for context cues.** (**A**) Context cues function to gate the routing of subsequent target cues. Target cues in this study are reward-predictive cues whose reward-predictive properties depend on the identity of the preceding context cue. Top: schematic of network activity for a pool of neurons in response to a series of motivationally relevant cues, similar to that presented in 1. In this schematic, a behavioral response to cue 1 is rewarded when preceded by context cue 1 but not by context cue 2, and vice versa for cue 2. After presentation of a context cue, the network shifts to a new activity state, characterized by a change in the firing rates of individual neurons, with each context cue triggering a distinct network state. Furthermore, the difference in the excitability of individual units after presentation of the context cue then determines the receptivity of individual units in the network to subsequent response of the target cue, allowing the generation of dynamic value estimates to facilitate the appropriate Go/No-go response. In this case, the network is coding a gating response that is modulated by the context. Bottom: hypothetical PETHs for each trial type for a context coding population-level representation. Note, the representation discriminates firing to the 2 context cues and this difference is sustained until target cue presentation (purple vs. orange lines). Red lines border presentation of the context cue; black lines border presentation of the target cue (see Fig 1 for further details). (**B**) Context cues are interpreted in the NAc by their proximity to a rewarded state. Top: Presentation of either context cue elicits a similar network state that is further amplified upon presentation of the rewarded target cue. This ramp-like activity in the network encodes (discounted) expected future reward, referred to as a "state value" in reinforcement learning. Bottom: hypothetical PETHs for a population-level state value coding signal. Note, the peak activity during presentation of the target cue for rewarded trial types (dark colors) but not unrewarded trial types (light colors), and the ramp leading up to this via the context cues. (**C**) Context cues are not dissociated from other motivationally relevant cues in the NAc. Top: Presentation of any of the separate motivationally relevant cues (context or target) elicits a similar network response in the NAc. The NAc is coding a general cue response. Bottom: hypothetical PETHs for general cue coding. Note, the example representation responds identically to all cues. NAc, nucleus accumbens; PETH, peri-event time histogram.

by shuffling the trial identity of each trial, while leaving the within-trial temporal dynamics unaltered (note, this process will preserve any general time-varying component that is independent of trial type). The strongest component across all animals was a nonspecific time-varying signal whose activity and time course was related to the time course of the odor cues, and, for this reason, we call it the "general cue" signal (Figs 5C and 7A; variance explained, M040: 23.9%; M111: 32.3%; M142: 34.1%; M146: 14.2%). Another strong nonspecific signal observed across mice was a ramping signal that increased in magnitude from context cue onset to after target cue onset (Figs 5B and 7B; variance explained, M040: 4.2%; M111: 4.7%; M142: 5.5%; M146: 4.5%). Furthermore, there was a significant positive linear relationship between this component and time (adjusted R-squared, M040: 0.90; M111: 0.93; M142: 0.91; M146: 0.79), that was substantial larger than the next closest component and time (next largest R-squared with a positive relationship M040: 032; M111: 0.60; M142: 0.23; M146: 0.48), consistent with what would be expected from a ramping "state value" signal.

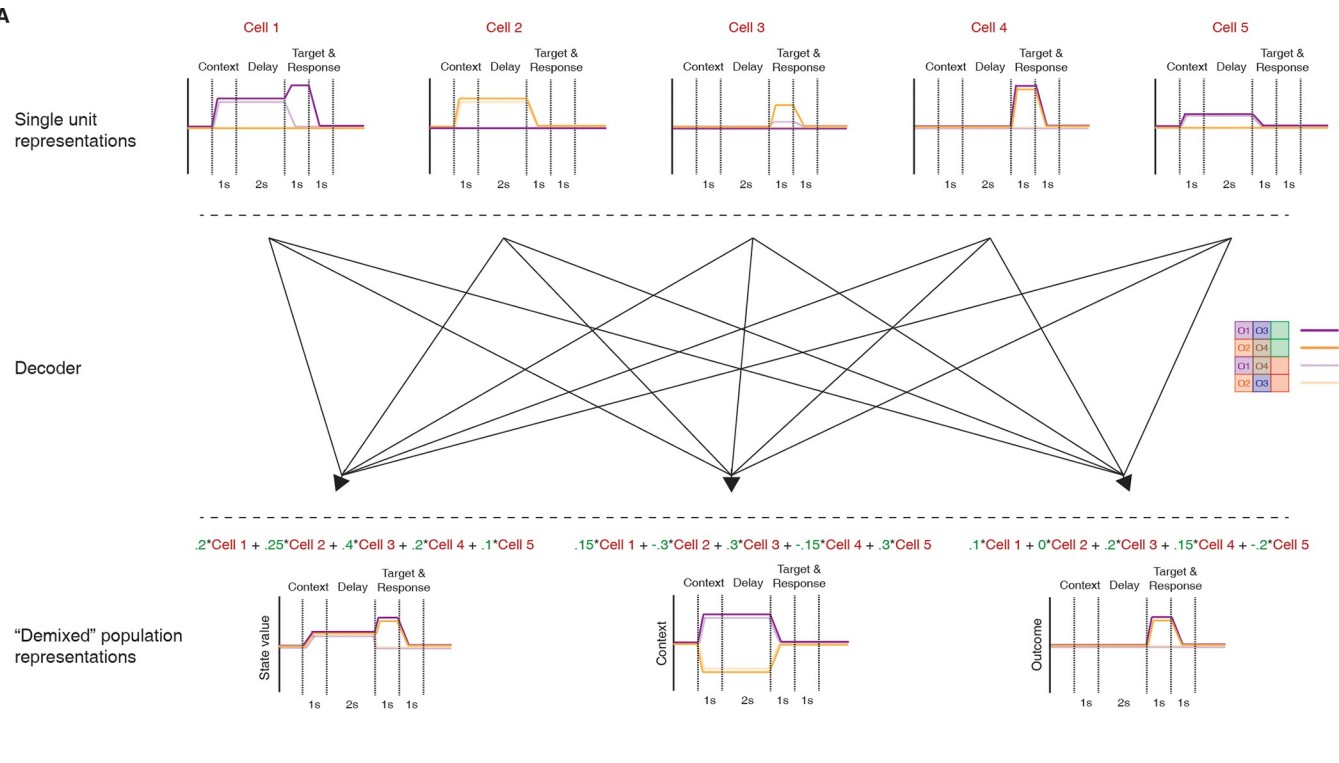

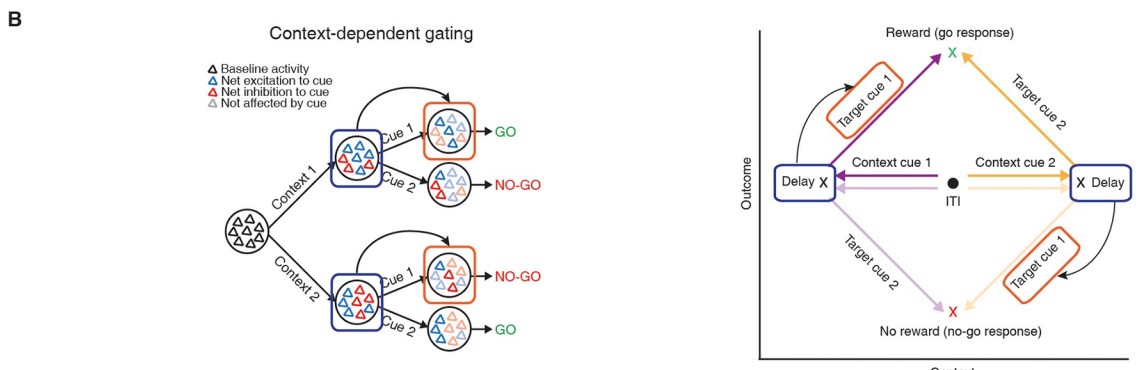

**Fig 6. Schematic of workflow for population-level analyses.** (**A**) To investigate population-level representations of task features, and their relationship to one another, requires a dimensionality reduction technique that can extract latent variables representing individual features of the task, while preserving the structure of the data. In dPCA, pseudo-ensemble activity for each mouse, shown here by 5 hypothetical single-unit response profiles (top), can be reduced into a few key behaviorally relevant components (bottom) through a decoder (middle) that seeks to minimize reconstruction error between reconstructed data and task-specific trial-averaged data. Note, there are multiple ways to combine individual units to generate population-level representations. In this example, orthogonal context- and outcome-related representations are extracted, suggesting that these 2 patterns of activity occupy separate subspaces in the neural activity space. Green numbers represent the weights of a unit for a given component. (**B**) The hypothetical context- and outcome-related components (right) can be used to test the feasibility of the context-dependent gating hypothesis (left). Shown on the bottom is the progression of neural activity through a trial for each trial type in a two-dimensional neural subspace, with the trial-averaged projected activity in the context-related component on the x-axis, and the trial-averaged projected activity in the outcome-related component on the y-axis. If the context-related component brings the network to a distinct state (note the separation along the context-axis from ITI to delay) that modulates the input–output mapping of the subsequent target cue (note the separate paths taken by a target cue in the neural space for each context cue), then a quantifiable relationship should exist between the 2 components at these time points. This can be tested by using linear regression to predict activity (arrow) in the value axis during target cue presentation (red box) from activity in a context axis during the delay period preceding target cue presentation (blue box). dPCA, demixed principal component analysis; ITI, intertrial interval.

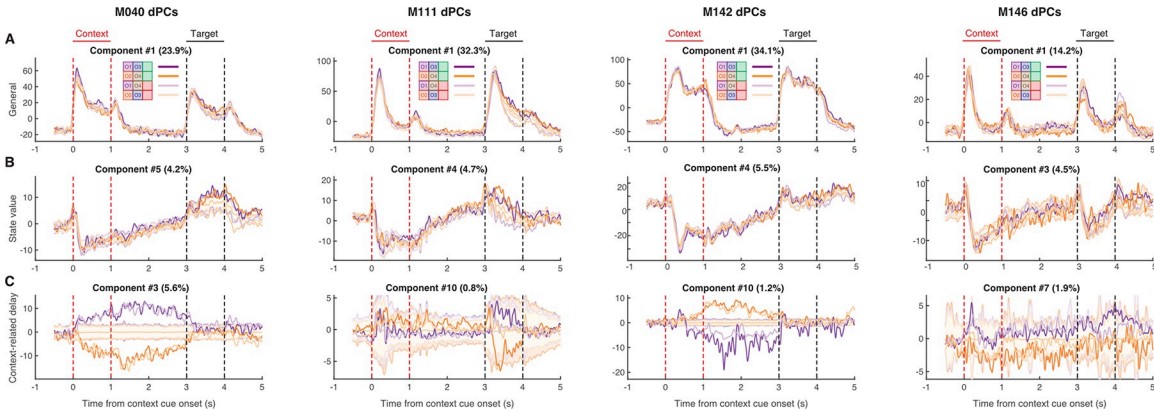

**Fig 7. Extracted components related to the hypothetical coding scenarios demonstrating the presence of coexisting signals during the context and delay period for each mouse, as outlined in Fig 5.** dPCA was applied to the pseudo-ensemble data from each mouse to extract low-dimensional population representations for various task features. Each plot represents the trial-averaged projected activity onto the component for a given task feature (rows) for each mouse (columns) for each trial type (purple: trials with context cue O1; orange: trials with context cue O2; dark colors: rewarded trials; light colors: unrewarded trials). Shaded regions and associated lines indicate 2.58 standard deviations ($p < 0.01$) and the mean of a shuffled distribution where trial identity was shuffled before trial-averaging the data. This shuffling procedure removes information about trial identity while preserving the temporal dynamics of condition-invariant signals. Plot title denotes the overall ranking of the component, and the amount of variance explained by the component. Red lines border context cue presentation, and black lines border target cue presentation. From left to right shows components for M040, M111, M142, and M146. (**A**) Top nonspecific signal that responded to all odors and called the general cue component (Fig 5C). Note that this signal is present during presentation of both context and target cues. (**B**) The extracted component from each mouse that best represents a state value signal (Fig 5B), with a ramping-like activity between context cue offset and target cue onset. Note the variability in this component across mice after target cue onset, in particular the separation between rewarded (dark colors) and unrewarded (light colors) trials for M040 and M111. (**C**) The context-related component that best separated context cues during the delay period (Fig 5A). Note that in the mice where this signal is strongest (M040, M142), a strong separation between context O1 (purple) and context O2 (orange) trials appears from context cue onset until target cue onset. Data: https://gin.g-node.org/jgmaz/BiconditionalOdor. dPCA, demixed principal component analysis.

In terms of components related to the context cues, there are several noteworthy observations. First, there was heterogeneity across mice in terms of the magnitude of the context and delay components (variance explained for 2 context-related components, M040: 7.0%; M111: 1.3%; M142: 2.8%; M146: 2.8%), closely following the observations from the single-unit data and pseudo-ensemble predictions. Second, in the mice with the largest degree of separation in context-related neural space, there were 2 distinct patterns of activity, a signal that was most dominant during the delay period that followed the context cue, the "delay" signal (Figs 5A and 7C), and a signal that followed the time course of the context cue, the "context" component (S3 Fig). The dot product between the delay context-related component and the general cue (M040: 0.07; $p = 0.136$; M111: 0.05; $p = 0.219$; M142: 0.18; $p < 0.001$; M146: 0.24; $p = 0.002$) and state value (M040: −0.07; $p = 0.156$; M111: 0.05; $p = 0.242$; M142: 0.04; $p = 0.259$; M146: 0.26; $p < 0.001$) components did not significantly deviate from zero in some, but not all, mice, suggesting that in some cases, these signals are orthogonal and can coexist independently within the same population of NAc units. Finally, there were also clear components related to the identity of the target cue, the "target" component (S3 Fig), and the behavioral response of the animal, the "outcome" component (S3 Fig), which were consistently orthogonal from the delay context-related signals (M040: 0.03; $p = 0.340$; M111: −0.03; $p = 0.340$; M142: 0.04; $p = 0.257$; M146: 0.03; $p = 0.359$). Together, this suggests that all aspects of task-related activity are represented in the population-level activity of the mice, including clear state value representations that were not generally apparent in the single-unit responses. Furthermore, unlike the single-unit analysis, the presence of distinct population-level representations within a single neural population opens up second-order questions that allow the

investigation of how individual components are related to each other within the same neuronal population, such as testing for context-dependent gating of outcome-related representations, which we do below.

To test which of these population-level representations are responsible for the previously demonstrated capacity of the pseudo-ensemble activity to accurately predict the behavioral response for a target cue, we ran the same binomial regression, predicting lick or no lick for a given target cue, using the pseudo-ensemble activity from each mouse projected onto the extracted population-level components for each task feature (S4 Fig). Only the context-related components were able to predict information about the animal's upcoming response to each target cue during the delay period, and the strength of this prediction was related to how strong the component was in a given mouse (prediction accuracy for the context-related delay component at the end of the delay period, M040: 99%; z-score: 3.06; $p = 0.002$; M111: 62%; z-score: 1.69; $p = 0.091$; M142: 92%; z-score 3.98; $p < 0.001$; M146: 83%; z-score: 3.38; $p < 0.001$). In fact, in the 2 mice that showed the largest separation in the delay context-component, the predictions using this component alone surpassed that of the entire pseudo-ensemble, suggesting that this context-related activity is related to the animal's subsequent response. Furthermore, neither the general cue or state value components contained this predictive information during the delay period (S4A and S4B Fig).

## Context-specific ensemble states predict the magnitude of the subsequent outcome-related response

The findings of clear context-related components that hold information about the trial context during the delay period until presentation of the target cue, and the ability of these components to predict the animal's subsequent response, raises the possibility that this activity might be able to predict the behavioral response via gating behaviorally relevant activity to the target cue. In this task, behaviorally relevant activity could represent a variety of task-related variables, such as the expected value of the outcome predicted by the target cue, or the optimal action to take. Note, however, that our experimental design does not dissociate these different kinds of behaviorally relevant signals (see Discussion). If each context cue brings the NAc to a unique network state that possesses distinct input–output transformations for a given target cue to enable the generation of context-appropriate outcome-related representations (Fig 5A), then a biomarker for this relationship should be observed in the linking between context-related and outcome-related population representations. For instance, if the transformation of target cue "O3 into a behaviorally relevant representation is dependent upon whether the NAc is in the network state related to context cue "O1 or context cue "O2, then context-related activity during the context period should be informative of subsequent outcome-related activity to the target cue (see Fig 6B for schematic; Fig 8A, S5–S7 Figs for data trajectories across mice). If, on the other hand, the context-related network state is not relevant for the transformation of the target cue into a behaviorally relevant representation, then there should be no relationship between the context-related and outcome-related components. To test this, we trained a linear regression to predict the activity of the top outcome-related component during target cue presentation, from the delay context-related component, and found that the delay component could account for a significant amount of variability for the outcome-related component for a given target cue during the delay period (Fig 8C for M040, S5–S7 Figs for M111, M142, and M146; proportion of variance explained at end of delay period, M040: 0.45; z-score: 63.16; $p < 0.001$; M111: 0.15; z-score: 7.25; $p < 0.001$; M142: 0.42; z-score: 10.92; $p < 0.001$; M146: 0.22; z-score: 8.89; $p < 0.001$). As a control, we also tried to predict variability in the outcome-related component to a given context cue but were generally unable to (Fig 8D for

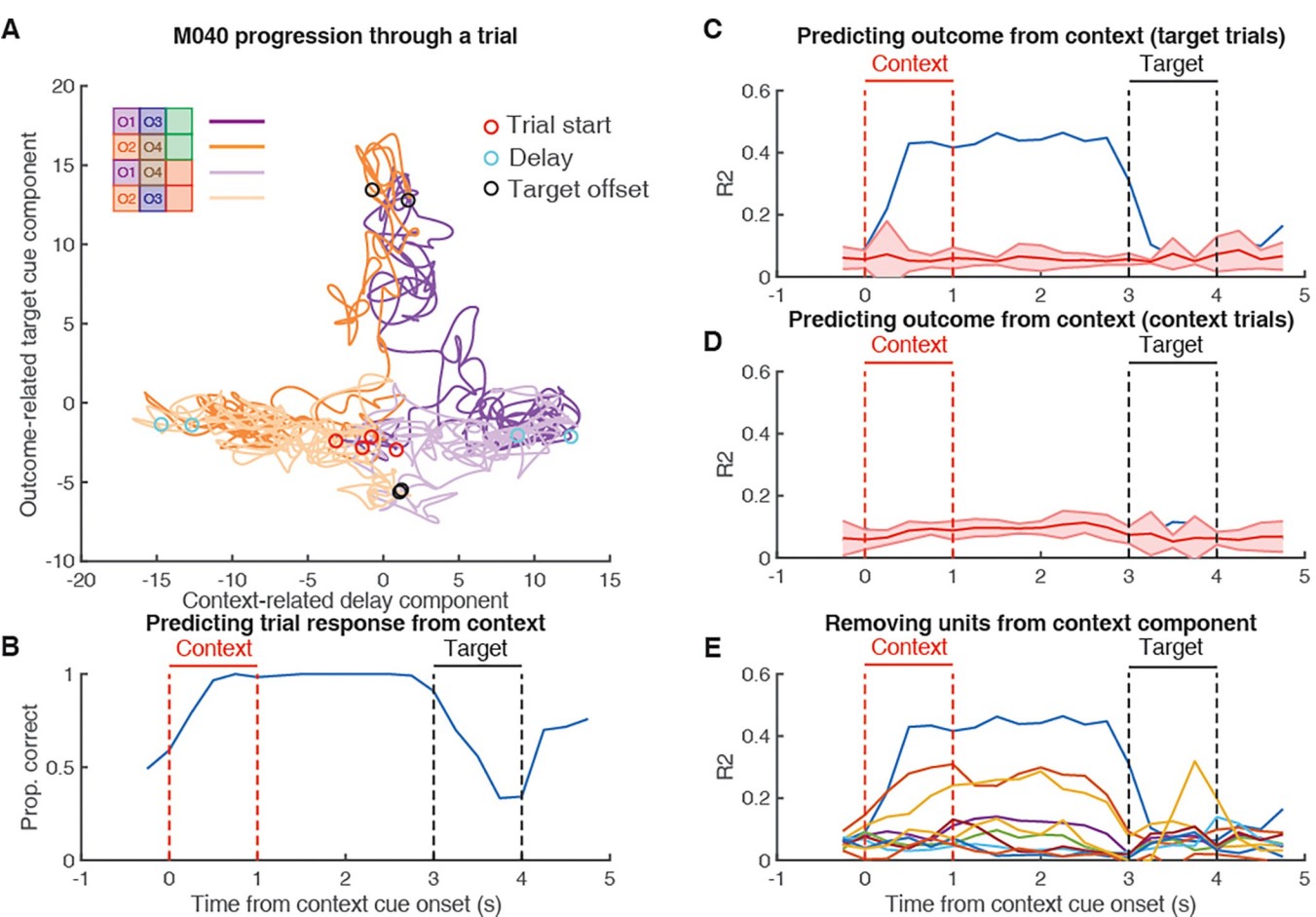

**Fig 8. Context-dependent gating of the outcome-related component, as outlined in Fig 6B.** Shown is the relationship between the context-related delay component and the outcome-related target cue component for M040, data for M111, M142, and M146 can be found in S5–S7 Figs. (**A**) Progression of neural activity through a trial for each trial type (purple: trials with context cue O1; orange: trials with context cue O2; dark colors: rewarded trials; light colors: unrewarded trials) in a two-dimensional neural subspace, with the trial-averaged projected activity in the context-related delay component (see Fig 7C) on the x-axis, and the trial-averaged projected activity in the outcome-related component (see S3 Fig) on the y-axis. During the context and delay periods a separation is observed along the context axis between context O1 and context O2 trials, after which separation is observed along the outcome-related axis following target cue presentation for rewarded versus unrewarded trials. Red circles signal context cue onset; cyan circles signal delay period 1 s after context cue offset; black circles signal 1 s after target cue onset. (**B**) Using a binomial regression to predict the behavioral response (lick or no lick) for a given target cue based on projected activity along the context-related delay component at various time points, showing the high accuracy during the context and delay periods. Red lines border context cue presentation, and black lines border target cue presentation. (**C**) Using a linear regression to predict projected activity along the outcome-related axis after target cue onset for a given target cue (black circles from A) based on projected activity in the context-related axis at various time points, showing performance above chance levels during the context and delay period. (**D**) Control analysis showing the inability of using a linear regression to predict projected activity along the outcome-related axis after target cue onset for a given context cue based on projected activity in the context-related axis. (**E**) Iteratively removing the top 10% of contributors to the context-related delay component and repeating the linear regression-based analysis of predicting outcome-related activity as in C, showing the ability to achieve above chance perform even after removing the top 20% of single-unit contributors to the context-related delay component. Data: https://gin.g-node.org/jgmaz/BiconditionalOdor.

M040, S5–S7 Figs for M111, M142, and M146; proportion of variance explained at end of delay period, M040: 0.09; z-score: 0.08; $p = 0.936$; M111: 0.13; z-score: 1.59; $p = 0.112$; M142: 0.21; z-score: $-1.56$; $p = 0.119$; M146: 0.11; z-score: 0.14; $p = 0.889$). This suggests that the ability of the delay context-related component to predict variability in the outcome-related component was specific to the delay component and not a general feature of the population-level representations. Furthermore, to assess whether this effect was driven by a few highly contributing units, we repeated these predictions while iteratively removing the top 10% of contributors to the component and found that this prediction persisted even after removing the top

20% of units (Fig 8E for M040, S5–S7 Figs for M111, M142, and M146). Together, these findings suggest that the NAc ensemble encodes a context-related component during the delay period and that this activity is linked to subsequent behaviorally relevant coding during the target cue period in a way that supports the context-dependent gating account of context coding. Importantly, this gating feature was unique to the population-level representations and was not apparent from analysis of single-unit activity.

## Discussion

A dominant view of NAc function and what is encoded in its activity is that it tracks fundamentally value-centric quantities, such as incentive salience, value of work, expected future reward, economic value, risk and reward prediction error [1,6–9]. Here, we present a number of findings that demonstrate this view is too narrow. In particular, we demonstrate that NAc single-unit (26% of units across mice) and population-level representations distinguish between 2 context cues in a biconditional discrimination task. This effect is not likely due to unequal cue salience across context cues, because we counterbalanced the odor associations across mice. It is also unlikely to be a result of associations between the context cues and reward, because we observed behavioral performance (and therefore the amount of reward obtained) to be similar across context cues for each animal. Importantly, at both the single-unit and population level, context coding persisted throughout the delay period when the animal must maintain a representation of this cue to inform subsequent behavior to the target cue, demonstrating it is not simply a sensory response. Additionally, at both the single-unit and population level, activity during the context and delay period could predict the subsequent behavioral and neural response for a given target cue and contained statistically independent components encoding a nonspecific cue signal, a ramping signal, and a context signal. To our knowledge, this study is the first to show in rodents that NAc units discriminate between context cues that are not directly tied to reward, but instead set the expected value of subsequently presented reward-predictive cues. This context signal has properties suitable for the neural implementation of important NAc functions such as gating the current behavioral relevance of such cues, and/or in appropriate credit assignment based on feedback, as we discuss below.

### Context coding in the NAc

We modeled context in our study by the simplest possible implementation: a discrete cue that signals distinct task states with separate stimulus–outcome associations for subsequently presented stimuli [57,58]. In general, the term "context" can refer to any circumstance that changes the meaning of a specific target stimulus [59], as illustrated by the behavior of our subjects in emitting a different response (lick, no-lick) to a given target cue depending on the identity of the preceding context cue. However, this definition does not specify the particular process or mechanism through which the context cue comes to modify the response to the target. Two prominent, nonexclusive possibilities are (1) the context modifies the association between the CS and US ("occasion-setting") and (2) the context forms a configural cue with the CS, which then gets associated with the US. In addition, (3) the context cues may enter into direct association with the US themselves [60,61]. Because we observed no licking CR in response to the context cues, (3) seems unlikely to be a major contribution. Although our experimental design incorporated a delay between the presentation of the context and target cues, impairing the ability of the context cue to enter into a configure with the target, we cannot entirely rule out the possibility that some neural trace of the context cue remains after its offset (see the discussion in the next paragraph). Thus, our experiment does not allow us to determine the precise contributions of (1) and (2), which could be addressed in future work

with behavioral experiments such as extinction and transfer tests, and perhaps representational similarity analyses on the neural data that compare across elemental and configural cues. Importantly, our conclusions in this study do not depend on (1) or (2) being the case; either way, our data demonstrate the existence of a context signal in the functionally important sense that it is value independent and predictive of the subsequent response to a given target cue.

What is the nature of the context signal in the NAc? Our results show that the activity of a significant proportion of single units (26% across mice) discriminates between the 2 context cues, but this finding does not by itself specify the information encoded by this signal. In particular, since we only used 1 cue per context, the identity of that cue itself could serve as the context signal. While we believe that the 2 s of active flushing of the odorant during the delay between cues is sufficient for the context cue odor to disperse from the experimental apparatus, previous work suggests that the olfactory bulb maintains an "after-image" of the odorant [62], and thus we cannot exclude the possibility that a representation of the context odor remained and combined with the target cue to form a configural cue. However, we think this is unlikely to be a complete account of the data, given the presence of single-unit and population-level correlates that only discriminated between context cues during the delay period following context cue offset. A different possibility is that the context signal encodes an abstract task state independent of the sensory properties of the context cue(s); this idea could be probed by having multiple distinct cues signal the same context and testing whether cue identity or context identity is a better match for the neural response. However, in either case, the signal retains the functionally important properties of a context in modifying the subsequent response to a given target cue.

Although each individual animal had individual units that showed context coding, there was also heterogeneity in the degree of context coding across mice. The clearest relationship between this heterogeneity and any other component of the experiment was in the training duration for the animals, with mice that took fewer days to reach performance criterion on the task before recording showing less separation in the population-level context component, while those that took more days to reach criterion had a larger separation in the context component. Given the small sample size in each group ($n = 2$), it is hard to draw any substantial inferences from this data. However, speculatively this finding may suggest that during initial learning, the NAc receives cue information and processes it solely in terms of its motivational relevance, and some other structure is supporting context-dependent behavior, but then after extensive learning, it forms representations of the associative structure. Interestingly, recent work has suggested that motivated approach behavior becomes less NAc dependent as training progresses [63]. Whether or not this is related to a shift in the role of the NAc as a behavior becomes learned, or related to distinct inputs such as from the hippocampus and cortex, remains to be determined. An additional potential contributor to the variability across mice is recording location. The mouse that contained the lowest number of context-related single-unit responses was also the mouse whose recording coordinates spanned the most caudal aspect of the NAc. Several lines of investigation suggest a heterogeneity in NAc processing along the rostral-caudal gradient [64–67], largely due to a different distribution of inputs, suggesting that a portion of the observed variability in context coding may be due to differences in recording location.

The present study expands upon our previous work demontrating that NAc units distinguish between different conditioned stimuli of equal reward-predictive value (lights and sounds [46]). Interestingly, a subset of these stimulus set-discriminating units also showed sustained changes in firing during trial periods *before* the presentation of the cue, suggesting that they encoded an abstract task feature not directly tied to stimulus presentation. However, because in this study cues were presented in blocks and did not modify the meaning of

subsequently presented cues, we could not determine the precise nature of this signal. The present experiment addresses these prior limitations by being the first NAc recording study to present both distinct, temporally precise context cues, and cues with dynamic outcome-predictive properties, demonstrating that the NAc codes information about the context that continues into the delay period. Additionally, our work is comparable to previous work that found evidence for rule encoding in the primate NAc [45], suggesting that this rule coding might be part of a more general NAc computation that primes the network to distinct states to enable behavioral flexibility. Together, our study is the first to demonstrate the presence of context-related correlates in the rodent NAc, providing further support for the burgeoning recognition of the NAc in decision-making outside of reward processing.

### Relationship between context and behaviorally relevant coding in the NAc

As discussed above, a salient functional requirement of a context signal in the brain is that it should have the ability to modify the response to a given target stimulus: in our task, given context odor 1, the correct response to odor 3 is to lick for reward, but given context odor 2, the correct response is to withhold. Throughout the paper, we refer to target cue activity that is related to the interaction between context and target cues as "outcome-related or "behaviorally relevant activity, to signify activity that cannot be explained by the identity of the target cue. Given that the value of a trial and the behavior of the animal were correlated in our task, that is rewarded trials were followed by licking behavior, and unrewarded trials were followed by the absence of licking, we cannot fully separate the contributions of subjective value and subsequent behavior in our behaviorally relevant components. Future work implementing an explicit delay between presentation of the target cue and the subsequent behavioral response would enable this dissociation. However, this confound does not interfere with the interpretation of our primary finding of context coding, or the linking of context-related activity during the delay period with subsequent, behaviorally relevant activity during the target cue period.

In addition to showing context coding using a biconditional task design specifically designed to control for value, a further innovation in this study is the population-level analysis, which allowed us to show evidence for coexisting activity patterns in the population-level representations, as well as a functional link between context coding and subsequent processing of the target cue. To determine whether our behavioral predictive power was arbitrary to any population-level component, we ran the binomial regression on all major components and found that only those containing significant context information had predictive utility during the delay period. These population-level results align with a growing body of work advocating for population-level interpretations of neural data, suggesting that certain neural computations are better understood in terms of their population-level versus single-unit output [68–71]. A primary argument for this approach is that there is a high degree of correlation and redundancy across single-unit coding, suggesting that the large neural space occupied by units in a region can be captured by a drastically lower dimensional latent space. Furthermore, the utility of single-unit output is ultimately determined by how it is integrated with other inputs by a receiver network. A proxy for this integration can be assessed by investigating the individual unit weightings of components extracted from dimensionality reduction techniques, as the weights for a component hypothetically represent how units are integrated for a particular output signal. We next discuss interpretations of our findings through this population-level framework below.

The finding of a clear nonvalue signal supports other work that the NAc is coding for more than a low-dimensional value signal [45,46]. A potential function for the context coding observed in the present study is implementation of the hypothetical switchboard function of

the NAc [53,54], serving as a routing mechanism to enable dynamic value representations of target cues (gating; Fig 6B). Indeed, the distinct occupancy in the pseudo-ensemble space for each context cue signals that the context cues might be driving the NAc into separate network states, setting an initial state for subsequent input–output flow of the target cue. In addition to the presence of a context signal, this routing function would also require that the context signal is both functionally linked to the subsequent outcome signal, while simultaneously not interfering with outcome-related output. We found support for the former from the observation that activity in this space during the delay period could explain a significant proportion of variance in the behaviorally relevant component during the target cue period. Context-specific cortical input that modulates the excitability of individual NAc neurons, resulting in a differential response to the subsequent target cue, is a candidate mechanism for these population-level observations. Similar population-level observations have been observed in the field of motor control and, more recently, economic choice [72–74]. Furthermore, given that the context-related and behaviorally relevant components were orthogonal from one another, it suggests that context coding does not interfere with the ability of downstream structures to read-out outcome-related information. However, given that our experiments were correlative in nature, these interpretations are speculative, and future work is needed to test the causal contributions of these components. Finally, another potential functional role for the observed context coding is in forming the associations between reward-predictive cues and the rewards themselves. Recently, several studies have implicated cortico-striatal circuits in credit assignment [75,76], raising the possibility that this context signal may be used for learning to assign credit to the appropriate state. Whether these context components are generated in the NAc or inherited from inputs, as well as if they represent an internal computation that is locally used to organize NAc activity, or are conveyed downstream, remains to be determined.

Beyond context-dependent gating interpretations of the data, we also found support for reinforcement learning-inspired accounts of NAc function [10,55] (state value; Fig 5B). For instance, all mice showed a ramping component after context cue onset, consistent with dynamics that closely mimic what would be expected from a signal conveying state value. Interestingly, this signal coexisted within the same population of units as the context signal and was orthogonal from the delay context-related component in the 2 mice with the clearest rewarded versus unrewarded discrimination, suggesting that the NAc can process both types of information. This signal is similar to ramping signals observed previously in single-unit studies [13] and may be the result of the strong hippocampal input to the NAc. Future experiments inactivating hippocampal drive should test the relationship, as well as the necessity of this signal for proper evaluation of outcome-predictive cues.

Interestingly, across all the animals, the strongest extracted component was a general cue signal that signaled the onset and duration of all cues used in this study (general cue; Fig 5C). These condition-independent signals are being found across various domains of systems neuroscience [77–79], and their strikingly relative dominance to other task-related signals suggest that they signal general task-related transitions in neural state space, although their exact role is unknown. Given the NAc is part of a broader limbic network that is entrained by respiration and has strong connections with olfactory-processing regions, it is possible this component is signaling to the NAc the presence of a salient event and priming it to process the associative content of the cues, perhaps by increasing the excitability of the NAc and opening the afore-mentioned gate. Regardless of the precise functional relevance of the observed correlates in this study, the finding of clear nonvalue correlates suggests a revision of the value-centric account of neural activity, encouraging future work to view NAc activity as a richer signal containing more than just reward.

## Methods

### Subjects

A total of 4 adult female wild-type C57BL-6J mice (Jackson Labs) were used as subjects (data from a 5th mouse were collected but were not analyzed due to poor behavior). Male mice were also trained on the task but did not perform enough trials to make it to the recording stage of the experiment. Mice were group housed before being selected for the experiment with a 12/12-h light–dark cycle, were individually housed once training commenced, and tested during the light cycle. Mice were food restricted to 85% to 90% of their free feeding weight (weight range at start of experiment was 19.7 to 23.8 g) and water restricted for a minimum of 6 h before testing. All experimental procedures were approved by the Dartmouth College Institutional Animal Care and Use Committee (IACUC; protocol #00002171) and carried out in accordance with the National Institute of Health's Guide for the Care and Use of Laboratory Animals.

### Overall timeline

Mice initially underwent surgery where craniotomies were marked and a headbar was affixed to the skull. After a 3-d recovery period, mice were food restricted and acclimated to being handled by the experimenter and being held by the headbar in the experimental room for 3 d. Mice were then habituated to being head-fixed on the apparatus over the course of 3 or more days, starting with 5 min and working up to an hour of being head-fixed. During later headbar habituation sessions, animals were placed on water restriction and trained to lick a spout for 12% sucrose solution. After learning to lick for sucrose (1 to 2 sessions), mice were then trained to lick in response to the rewarded odors in the task for 1 to 3 d before undergoing full task training. Once behavioral criterion was reached in the full task (6+ d; described below), the first craniotomy was made, and acute recordings commenced after a 24-h recovery period. Recording sessions were carried out for 5 to 7 d, after which a contralateral craniotomy was made and the process repeated. After sufficient data collection, mice were killed and histology was performed to confirm recording sites.

### Behavioral task and training

In order to assess whether the NAc codes for information related to context cues, mice were trained to perform a biconditional discrimination task where they were presented with 2 odors in sequence. The identity of the first odor, the context cue, determined the value of the subsequently presented odor, the target cue, such that each context cue had a rewarded and unrewarded target cue pairing (Fig 1A and 1B; adapted from [49]). The apparatus was a custom built head-fixed mouse behavioral setup, consisting of a running wheel, odor port, lickometer, and headbar holders (Grasshopper Machine Werks LLC). Pressurized air passed through an olfactory delivery system containing 5 distinct tubes, with 1 tube containing mineral oil and the rest a mixture of mineral oil and a specific odorant. Each tube was connected to an experimentally controlled valve that then sent the air-odorant mixture to the odor manifold on the head-fixed setup, where the active line was sent to the mouse. Apart from odorant presentation, mice were continuously presented with unscented air via the mineral oil only line. Odorants used in the study were propyl formate, 1-butanol, propyl acetate, and 3-methyl-2-buten-1-ol (Sigma). Odorants were selected based on previous work using this task [49–51]. The lickometer detected changes in capacitance from mouse licks and sent this information to a Digital Lynx acquisition system (Neuralynx). The task was controlled via a custom-written MATLAB

script (Mathworks) that triggered TTL pulses from the acquisition system to control the odorant and sucrose valves.

After initial handling and habituation mice were first trained to lick a spout for 12% sucrose solution, until they manually triggered over 100 sucrose water rewards in <20 min. Mice were then shaped to lick in response to pseudo-randomized presentation of the 2 rewarded context–target odor pairs, with the context odor determining the outcome predicted by the subsequent target odor. Odor selection and pairing was pseudo-randomized across mice to ensure unique pairings across animals. A single trial consisted of a 1-s presentation of the context odor, followed by a 2-s delay where unscented air was presented to flush out the odorant, followed by a 1-s presentation of the target odor, followed by an additional 1-s response window, followed by a 12 +/− 2 s intertrial interval (Fig 1A). Licking either during presentation of the target odor or the subsequent response window registered as a correct response. During the shaping phase, sucrose water was delivered pseudo-randomly in 1/3 of the trials in which mice failed to lick. Mice were allowed to complete up to 200 trials in a session, with an individual session being terminated either at 200 trials or if the mouse became sufficiently disengaged by the task, measured by the absence of licking for 10 consecutive trials. This phase of training continued until the mouse licked for approximately 80% of trials. After the shaping phase, mice then underwent the full task training, where they were presented in pseudo-randomized sequence all 4 context–target odor pairs. Upon reaching criterion of 3 consecutive sessions with >80% correct responses (range: 7 to 28 sessions), a craniotomy was made over the first hemisphere, and recordings began after a recovery period.

## Surgery

Mice underwent 3 surgeries over the course of the experiment. The first surgery consisted of exposing the skull, marking the location of future craniotomies, and securing the headbar. The second surgery consisted of making the first craniotomy and installing a posterior reference wire above the cerebellum. The third surgery consisted of making the second craniotomy. In all surgeries, mice were anesthetized with isoflurane, induced with 5% in medical grade oxygen and maintained at 2% throughout the surgery (0.8 L/min), and were administered ketoprofen as an analgesic prior to surgery, with a supplementary dose 24 h after the procedure.

## Data acquisition and preprocessing

For recording sessions, 32 (NeuroNexus; A4x2-tet) or 64 (Cambridge NeuroTech; P-1) channel silicon probes were lowered into the NAc (AP: 0.8 to 1.4 mm; ML: +/− 1.0 to 2.0 mm; DV: 4.0+ mm). After letting the probes settle for 30 min, single-unit activity was recorded during behavioral performance. NAc signals were acquired using a Digital Lynx data acquisition system with an HS-36 PTB preamplifier (Neuralynx). Putative spikes were recorded as threshold crossings of 600 to 6,000 Hz band-pass filtered data with waveforms sampled at 30 kHz. Signals were referenced locally to maximize signal to noise of the spiking waveforms. Spike waveforms were clustered with KlustaKwik using energy and the first derivative of energy as features, and subsequently manually sorted using MClust (MClust 3.5, A.D. Redish). Isolated units containing less than 200 spikes during the trial period were excluded from analysis.

## Data analysis

**Behavior.**   If mice learned the appropriate associations between context and target cues, then correct behavior on the task would look like a high licking response rate to context–target pairings that are rewarded, and a low licking response rate to context–target pairings that are

unrewarded (see Fig 1B for hypothetical learned trial structure). To assess whether mice learned to discriminate between rewarded and unrewarded odor pairs, we compared the mean proportion of rewarded and unrewarded trials that the animal made a lick response for a given odor, relative to shuffling the trial type label for the mean proportion of trials licked for a given session. Furthermore, to assess whether mice were not responding differently to individual target cues, we also compared the mean proportion of trials with a lick for each target cue, relative to shuffling target cue identity for the mean proportion of trials licked for a given session.

**Single-unit coding.** To address our question of how the NAc responds to context cues and their relationship to target cues, we compared the mean firing rates for different trial types at different time points for each unit. To determine whether or not a unit responded at all to any context cue, we compared the 1-s pre- and 1-s postcontext cue onset period. To determine whether or not a unit discriminated between the context cues, we compared the mean firing rate for the 1-s presentation of each context cue. To determine whether or not a unit discriminated between the context cues during the delay period, we compared the mean firing rate for each context cue for the 1-s period preceding target cue presentation. Likewise, a similar comparison was performed for general target cue responsiveness (1-s pre- versus posttarget cue onset), target cue selectivity (target cue 1 versus target cue 2 during 1-s target cue presentation period), and outcome-predictive selectivity (rewarded versus unrewarded trials during 1-s target cue presentation period). All firing rate comparisons were related to a shuffled distribution where the trial identity was shuffled across trials. Overlapping proportions were determined to be significant if they were larger than shuffling the identity of significant units for each task parameter. For all analyses, a value of $+/- 2.58$ z-scores from the shuffled distribution was used as the threshold value for significance ($p < 0.01$). All analyses were completed in MATLAB 2018a. For single-unit examples, peri-event time histograms (PETHs) were generated by smoothing the trial-averaged data with a Gaussian kernel ($\sigma$: 100 ms).

To determine how context is signaled throughout the period between context cue onset and target cue onset, we calculated the proportion of units that showed discrimination in firing to the context cues at each time point in the trial in 0.5-s intervals. To determine whether this coding had any utility in informing future behavior, we used a binomial regression to predict the animal's behavioral response for a given target based on the firing rate for a unit.

**Population-level coding.** To assess population-level predictions of behavior, we generated pseudo-ensembles for each mouse from the data recorded across sessions. We then trained a binomial regression to predict the behavioral response for each target cue using the firing rates of the pseudo-ensembles. To determine how context cues are represented at the network level, we performed dPCA to extract information related to the context and target cue from the population of recorded units (see [80] for a detailed description of the methodology). dPCA is an analysis that aims to explain most of the variance in the data as in PCA, while also separating the data for several task parameters similar to how LDA does for a single task parameter (Fig 6A). The dPCA method first took the mean-subtracted, trial-averaged data for all units, and decomposed the population data matrix into a sum of separate data matrices that each represented the contributions of a different aspect of the task, and noise. These task features are inputs to the analysis set by the experimenter, and in the current experiment, the task inputs were context cue type, target cue type, and the interaction between context and target cue signifying cue value. The loss function of dPCA then used the ordinary least squares solution to find the transformation that minimized the reconstruction error between the reconstructed data and the deconstructed data, with the deconstructed data matrix representing the contributions of a given task parameter to the full trial-averaged data. Dimensionality reduction was then achieved via eigendecomposition of the covariance matrix of the transformed data, and the top components were stored. The explained variance of each component was the

fraction of the total variance in the trial-averaged data that could be attributed by the reconstructed data for that component.

In our task, we sought to "demix" the contributions of the context cue, target cue, and their interaction (e.g., "outcome-related") across time, projecting the data using components derived from these task variables, and visualized how the projected neural trajectories evolved throughout a trial in this reduced dPCA space (see Fig 6B for hypothetical trajectories along context and outcome axes). We then visualized the differences across trial types in these components via comparison to a shuffled distribution that shuffled the trial identity of each trial, while preserving the temporal dynamics, preserving condition-invariant signals. This analysis requires a sample size of 100 neurons to achieve satisfactory demixing, and thus, sessions within a mouse were pooled together to run on pseudo-ensembles. To identify the ramping "state value" component, we implemented a linear regression predicting the condition-averaged projected activity from each component with the time between context cue offset and

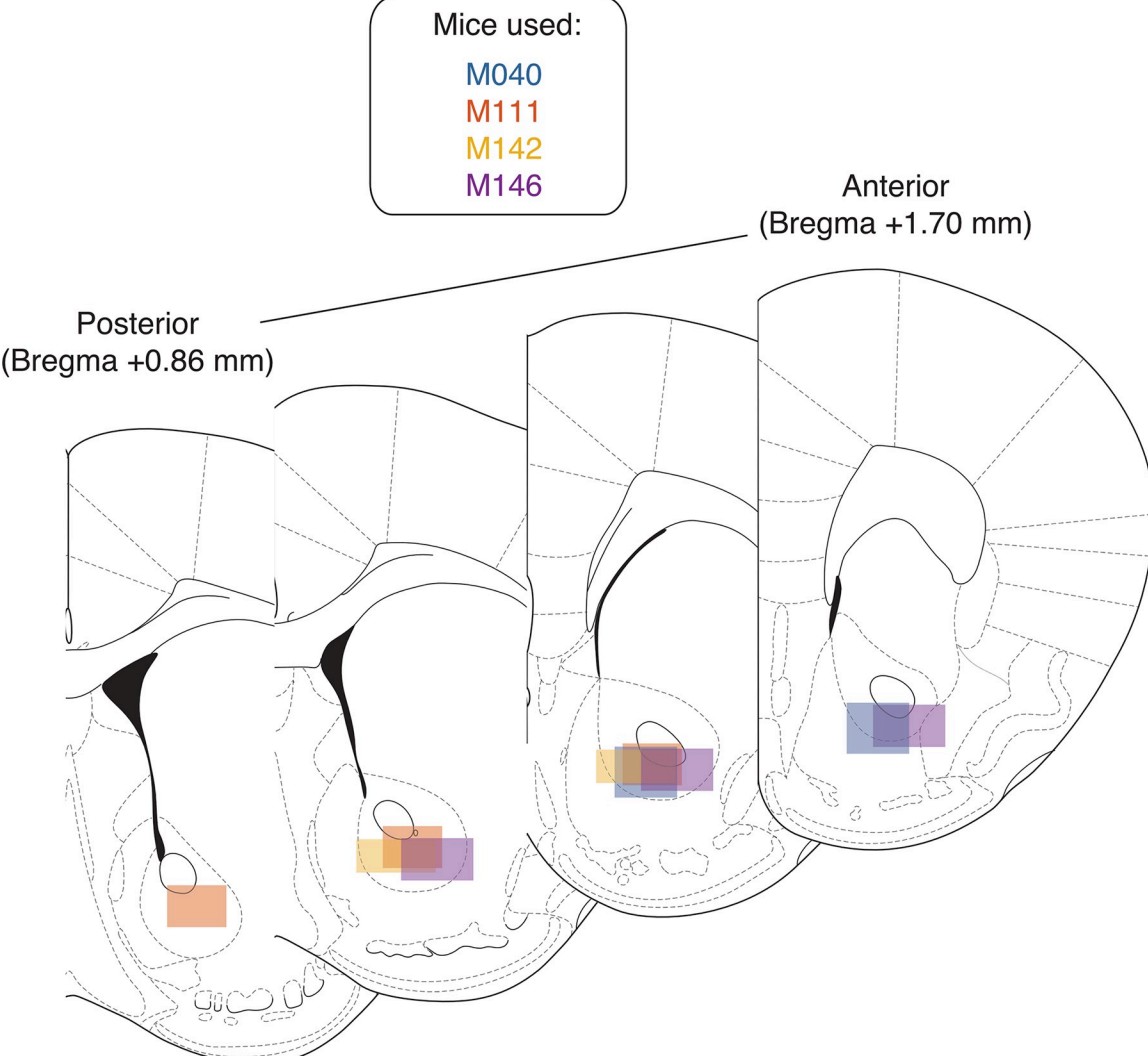

**Fig 9. Histology.** Schematic showing recording areas for all subjects, as determined by reconstruction of probe tracks following completion of neural recordings. Recordings were primarily obtained from the core of the NAc, but also included cells from the overlying caudate/putamen and the NAc shell. NAc, nucleus accumbens.

target cue onset. Furthermore, given that dPCA does not constrain the components extracted for each parameter to be orthogonal, components were identified as being nonorthogonal if the dot product significantly deviated from zero.

If activity in response to the context cue was indeed constraining subsequent information flow in response to the target cue, then we would expect to be able to predict both behavioral and neural features during the target cue epoch, based on the assumption that a given target cue would possess distinct input–output mappings for each context cue. First, to determine the contributions of the extracted components in predicting subsequent behavioral response to a target cue, we used binomial regressions to predict behavioral response from the projected activity using each component. Next, to more directly test the feasibility of our hypothesis of context-dependent gating of behaviorally relevant representations, we used a linear regression to predict the projected activity in the top outcome-related component during a particular target cue from the projected activity in the delay component across the 2 contexts. As a control, we also attempted to predict activity across target cues for the same context cue. Additionally, to determine whether any effects were driven by a few top contributors to these components, we repeated this analysis while iteratively removing the top 10% of units that had the largest weights.

## Histology

Immediately following the final recording session, mice were anesthetized with isoflurane, asphyxiated with carbon dioxide, and transcardial perfusions were performed. Brains were fixed and removed, and then sectioned in 50-mm coronal sections. Sections were then stained with thionin and visualized under light microscopy to determine probe placement (Fig 9).

## Supporting information

**S1 Fig. Average rate of licking during correct trials for each mouse across all recording sessions.** Top of each plot shows lick rasters during correct trials for the 4 trial types for all recording sessions (purple: trials with context cue O1; orange: trials with context cue O2; dark colors: rewarded trials; light colors: unrewarded trials). Bottom half of each plot shows trial-averaged licking rates for each trial type aligned to context cue onset. Data shown are averaged across all recording sessions. Context cue presentation (0–1 s) is bordered by red lines, and target cue presentation (3–4 s) is bordered by black lines. Note that for correct trials, substantial licking only appears after target cue onset, and for rewarded trials only. Data: https://gin.g-node.org/jgmaz/BiconditionalOdor.
(TIF)

**S2 Fig. Characterization of single-unit responses to the task for each individual mouse.** Each plot is a heat plot showing either max normalized firing rates or firing rate differences for trial-averaged data for all eligible units, with unit identity sorted according to the peak value for the comparison of interest. From left to right shows data for M040, M111, M142, and M146. Red lines border context cue presentation, and black lines border target cue presentation. (**A**) Firing rate profiles for units at 1-s pre- and postcontext cue onset, sorted according to maximum value after context cue onset. (**B**) Firing rate differences for units across context cues, sorted according to maximum difference during context cue presentation. (**C**) Firing rate differences for units across context cues during the delay period, sorted according to maximum difference during the 1-s period preceding target cue presentation. (**D**) Firing rate differences for units across target cues, sorted according to maximum difference during target cue presentation. (**E**) Firing rate differences for units for rewarded and unrewarded trial types

during target cue presentation, sorted according to maximum difference during target cue presentation. Data: https://gin.g-node.org/jgmaz/BiconditionalOdor.
(TIF)

**S3 Fig. Top behaviorally relevant components extracted for each mouse.** Each plot represents the trial-averaged projected activity onto the top 12 components (rows) for each mouse (columns) for each trial type. Plot title denotes the overall ranking of the component and the amount of variance explained by the component. Red lines border context cue presentation, and black lines border target cue presentation. From left to right shows components for M040, M111, M142, and M146. Components are ordered by amount of variance explained and include the following: a condition-invariant signal that responded to all odors ("general" cue component); a condition-invariant component present in most mice that showed a ramping-like activity after context cue onset, with a separation between rewarded and unrewarded trials after target cue onset ("state value" component); the context-related component that best separated context cues during the delay period ("delay (context)" component); the context-related component that best separated context cues during cue presentation ("context" component); the top target-related component that separated between target cues during target cue presentation ("target" component); and the top component that separated rewarded and unrewarded trials during target cue presentation ("outcome" component). Data: https://gin.g-node.org/jgmaz/BiconditionalOdor.
(TIFF)

**S4 Fig. Predicting behavioral response for a given target cue based on projected activity along the components extracted in Fig 7.** Each plot represents the accuracy of the behavioral prediction for a given component (rows) for each mouse (columns). Red lines border context cue presentation, and black lines border target cue presentation. From left to right shows predictions for M040, M111, M142, and M146. (**A**) Prediction accuracy for the general cue component. (**B**) Prediction accuracy for the state value component. (**C**) Prediction accuracy for the context-related delay component. (**D**) Prediction accuracy for the context component. (**E**) Prediction accuracy for the target component. (**F**) Prediction accuracy for the outcome-related target cue component. Data: https://gin.g-node.org/jgmaz/BiconditionalOdor.
(TIF)

**S5 Fig. Context-dependent gating of the outcome-related component.** Shown is the relationship between the context-related delay component and the outcome-related target cue component for M111. (**A**) Progression of neural activity through a trial for each trial type in a two-dimensional neural subspace, with the trial-averaged projected activity in the context-related delay component (see Fig 7C) on the x-axis, and the trial-averaged projected activity in the outcome-related component (see S3 Fig) on the y-axis. Note the relatively weak structure in the context-related delay axis, compared to M040. Red circles signal context cue onset; cyan circles signal delay period 1 s after context cue offset; black circles signal 1 s after target cue onset. (**B**) Predicting behavioral response for a given target cue based on projected activity along the context-related delay component at various time points. Red lines border context cue presentation, and black lines border target cue presentation. (**C**) Predicting projected activity along the outcome-related axis after target cue onset for a given target cue (black circles from A) based on projected activity in the context-related axis at various time points. (**D**) Control analysis predicting projected activity along the outcome-related axis after target cue onset for a given context cue based on projected activity in the context-related axis. (**E**) Iteratively removing the top 10% of contributors to the context-related delay component and attempting to predict outcome-related activity as in C. Data: https://gin.g-node.org/jgmaz/

BiconditionalOdor.
(TIF)

**S6 Fig. Context-dependent gating of the outcome-related component.** Shown is the relationship between the context-related delay component and the outcome-related target cue component for M142. (**A**) Progression of neural activity through a trial for each trial type in a two-dimensional neural subspace, with the trial-averaged projected activity in the context-related delay component (see Fig 7C) on the x-axis, and the trial-averaged projected activity in the outcome-related component (see S3 Fig) on the y-axis. Throughout the progression of a trial a separation is observed along the context axis, which then flows into the value axis after target cue presentation, similar to M040. Red circles signal context cue onset; cyan circles signal delay period 1 s after context cue offset; black circles signal 1 s after target cue onset. (**B**) Predicting behavioral response for a given target cue based on projected activity along the context-related delay component at various time points. Red lines border context cue presentation, and black lines border target cue presentation. (**C**) Predicting projected activity along the outcome-related axis after target cue onset for a given target cue (black circles from A) based on projected activity in the context-related axis at various time points. (**D**) Control analysis predicting projected activity along the outcome-related axis after target cue onset for a given context cue based on projected activity in the context-related axis. (**E**) Iteratively removing the top 10% of contributors to the context-related delay component and attempting to predict outcome-related activity as in C. Data: https://gin.g-node.org/jgmaz/BiconditionalOdor.
(TIF)

**S7 Fig. Context-dependent gating of the outcome-related component.** Shown is the relationship between the context-related delay component and the outcome-related target cue component for M146. (**A**) Progression of neural activity through a trial for each trial type in a two-dimensional neural subspace, with the trial-averaged projected activity in the context-related delay component (see Fig 7C) on the x-axis, and the trial-averaged projected activity in the outcome-related component (see S3 Fig) on the y-axis. Note the relatively weak structure in the context-related delay axis, compared to M040. Red circles signal context cue onset; cyan circles signal delay period 1 s after context cue offset; black circles signal 1 s after target cue onset. (**B**) Predicting behavioral response for a given target cue based on projected activity along the context-related delay component at various time points. Red lines border context cue presentation, and black lines border target cue presentation. (**C**) Predicting projected activity along the outcome-related axis after target cue onset for a given target cue (black circles from A) based on projected activity in the context-related axis at various time points. (**D**) Control analysis predicting projected activity along the outcome-related axis after target cue onset for a given context cue based on projected activity in the context-related axis. (**E**) Iteratively removing the top 10% of contributors to the context-related delay component and attempting to predict outcome-related activity as in C. Data: https://gin.g-node.org/jgmaz/BiconditionalOdor.
(TIF)

## Acknowledgments

We thank Andrew Alvarenga for the manufacturing of the lickometer and olfactory delivery system, and Jun Ho Lee for mouse husbandry advice.

## Author Contributions

**Conceptualization:** Jimmie M. Gmaz, Matthijs A. A. van der Meer.

**Data curation:** Jimmie M. Gmaz.

**Formal analysis:** Jimmie M. Gmaz.

**Funding acquisition:** Jimmie M. Gmaz, Matthijs A. A. van der Meer.

**Investigation:** Jimmie M. Gmaz.

**Methodology:** Jimmie M. Gmaz, Matthijs A. A. van der Meer.

**Supervision:** Matthijs A. A. van der Meer.

**Visualization:** Jimmie M. Gmaz.

**Writing – original draft:** Jimmie M. Gmaz, Matthijs A. A. van der Meer.

**Writing – review & editing:** Jimmie M. Gmaz, Matthijs A. A. van der Meer.

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
