## [Editor Report · Decision Letter 0]

19 Jun 2021

Dear Matt, 

Thank you for submitting your manuscript entitled "Contextual gating of motivationally-relevant stimuli in the mouse nucleus accumbens" for consideration as a Research Article by PLOS Biology.

Your manuscript has now been evaluated by the PLOS Biology editorial staff, as well as by an academic editor with relevant expertise, and I am writing to let you know that we would like to send your submission out for external peer review.

Please re-submit your manuscript within two working days, i.e. by Jun 22 2021 11:59PM.

Kind regards,

Gabriel Gasque

Senior Editor

PLOS Biology

ggasque@plos.org

---

## [Decision Letter · Decision Letter 1]

22 Jul 2021

Dear Dr van der Meer,

Thank you very much for submitting your manuscript "Contextual gating of motivationally-relevant stimuli in the mouse nucleus accumbens" for consideration as a Research Article at PLOS Biology. Your manuscript has been evaluated by the PLOS Biology editors, by an Academic Editor with relevant expertise, and by three independent reviewers.

In light of the reviews (below), we will not be able to accept the current version of the manuscript, but we would welcome re-submission of a much-revised version that takes into account the reviewers' comments. We cannot make any decision about publication until we have seen the revised manuscript and your response to the reviewers' comments. Your revised manuscript is also likely to be sent for further evaluation by the reviewers.

We expect to receive your revised manuscript within 3 months. 

**IMPORTANT - SUBMITTING YOUR REVISION**

Your revisions should address the specific points made by each reviewer, with new data and/or analyses where requested. 

Please submit the following files along with your revised manuscript:

*Re-submission Checklist*

*Published Peer Review*

*PLOS Data Policy*

*Blot and Gel Data Policy*

Sincerely,

Gabriel Gasque

Senior Editor

PLOS Biology

ggasque@plos.org

REVIEWS:

Reviewer #1: In this study, Gmaz and van der Meer recorded activity in the NAcc of head fixed mice while they executed a two-step odor-guided task in which an initial set of contextual cues (O1 or O2) indicated if a subsequent target cue (O3 or O4) would be rewarded or not. Since the NAcc has been traditionally (albeit not exclusively) linked to value coding, neuronal responses to the contextual cues should be similar, as they are associated with the same potential reward. However, the authors found a subset of NAcc neurons that discriminated between the contextual cues, i.e. they seemed to encode for abstract elements of task structure rather than just how much reward the mice should expect. These results implicate the NAcc in more complex cognitive representations than would have been expected, especially in mice.

It was a pleasure to read this manuscript. The hypothesis is relevant to the field, the findings are interesting, and the presentation is crisp and clear. However, I see a couple of issues that need to be addressed to support the author's conclusions.

Major comments:

- Studies with contextual cues typically employ cues of different sensory modalities (e.g. auditory/visual, visual/tactile, auditory/olfactory, etc.) to avoid potential compound cue representations. The authors only used odor cues, and partially addressed this potential confound. However, they argued as if this issue could arise only due to vestigial odor exposure during the delay. This is not the only concern, especially for odor cues. Transient exposure to odorants leads to persistent changes in firing in the olfactory bulb that can last for several breathing cycles (e.g. Patterson et al, PNAS 2013). It could very well be that the primary sensory representation of O3 and O4 is affected by persistent neural activity changes due to O1 or O2, and therefore the mice experience them as perceptually distinguishable cues. In effect, this would mean that the mice could be solving the task not by changing the value accrued to O3 and O4 based on the context, but rather by associating different percepts with the reward, i.e. O3'→ go, O3" → no go, O4'→ no go, O4"→ go. Persistent odor "after images" could explain the decoding of contextual cues during the delay period, and the fact that O3' and O3" would be distinguishable between themselves but still are more perceptually similar than O4' and O4" would explain the decoding of target odors from each other. Therefore, the arguments presented by the authors are insufficient to rule out this alternative explanation as unlikely. 

- Following the previous issue, given that there was just one pair of contextual cues and the analyses were all within subjects, and that it is possible than the mice are solving the task without necessarily using the "contextual cue" to actually gate context coding, then it is still also possible that the discrimination of O1 and O2 may reflect the sensory identity of the different cues. I agree with the authors that even if this is the case, it still means that NAcc neurons encode information that goes beyond value, which is the most poignant point of the study. But the authors should provide stronger arguments or data as to why their particular interpretation is more likely. 

Minor comments:

- It is very inconvenient to have to rotate the page to read figure 6. It would be best to rescale, or perhaps simplify, so it would fit in the standard portrait view.

- Is there any particular reason why only females were used? Are they easier to head fix? That is true for rats, but I wonder if it's the case for mice as well.

Reviewer #2: The manuscript by Gmaz et al. investigates how a non-value-based signal can predict the value of a subsequent cue-reward association through neuronal activity signatures in the nucleus accumbens. The authors designed a behavioral task in which mice receive a neutral cue that informs the rules of the task when subsequent cues are presented. Importantly, there is a delay period between the context cue and the onset of the cue, thus requiring a representation of this information to be prolonged. When they record single unit responses in the NAc to different aspects of this task, including what they term contextual cue coding, coding during the delay period, target cue coding, and value coding after target cue onset. The authors conclude that the NAc responds to various features of the task and can retain information about the neutral cue in the delay window before target cue onset. In addition, the authors looked at population-level activity and utilized dPCa to extract latent factors relating to the context cues, target cues, and trial value. They identified that the context signal extracted via dPCa is in fact predictive of the value encoded in the association between the target cue and reward. 

Overall, the data are interesting and informative and my enthusiasm for the data and results is high; however, there are several issues - some semantic and some experimental - that should be addressed before publication. The biggest issue I have with the manuscript is how the word value and non-value are used. The argument is that the cue that the authors term contextual does not have value; however, there is not data to show whether those cues do or do not have value. Further, while the authors call these cues contextual, I would argue that those cues serve as occasion setters - rather than contextual cues. 

Further, the task is entirely reward-based, so it is unclear if the responses to the reward itself are valence or other aspects of learning and behavior such as salience, which would not be able to be parsed in a purely reward-based task. 

Overall, the data are interesting, but the manuscript would benefit from more data showing that the contextual cue does not in fact acquire conditioned valence and more precision are care with speaking about the specific findings based on the data at hand. 

I have listed my specific comments below as major and minor. 

Major

1. The biggest weakness of the entire manuscript is the lack of behavioral data analysis showing whether the cue that the authors term contextual is actually a contextual cue (which would acquire value from the task) or an occasion setter (which by definition does not acquire value on its own). Regardless of whether the cue has intrinsic positive value because it signals that subsequently reward will be delivered, intrinsic negative value because it is a conditioned inhibitor of the incorrect response, or whether it is an occasion setter is irrelevant to the overall data, but is critical to the conclusions the authors are trying to make. Therefore, they should add behavioral data showing that 1. The animals learn this cue specifically and 2. Whether the cue does or does not have value itself. 

Here are some experiments that should be included - many of which only require reanalysis of the original behavioral data: 

o a raster plot of responses to the various trials would be appropriate to determine if mice are just making a couple of errant licks on the trials in which no reward is given or if there are actual lick bursts. 

o Do the mice primarily respond in the phase immediately following the target cue onset or after the 1 second presentation? I would also like to see inter-trial interval licks. Do the mice show any indiscriminate licking? Do they lick at all in the delay period?

o Show data to show that the mice to not lick in response to the first cue that is presented - i.e. there is no anticipatory licking and this cue has not acquired value. 

o It would also be important to show that the first cue is not a conditioned inhibitor of the secondary negative response. To do this you could present this cue concurrent with another CS+ and show that it does not reduce the rate of responding. 

2. In the paragraph following line 123, that authors state that the number context-coding units related to the number of training sessions. If you continued training sessions in mouse M111 until it reached at least 20 sessions, the authors would address whether the variability in context coding is in fact related to the number of training sessions rather than precise recording location or a factor of only needing so many context-coding units to maintain representation of the context during the delay. 

3. The manuscript suffers from an overall lack of precision in wording when talking about the data. For example, does value mean valence? The authors should go through and operationally define terms and use them appropriately. Along a similar line, I do not believe that the cues that the authors term contextual cues are actually contextual cues. Also the responses that the authors attribute to valence - i.e. reward responses - also may not be valence. In that situation many things are changing (reward prediction is met, salience is increased, valence goes up) and the experiments are not designed to parse these scenarios. The authors should take care to go through the manuscript to clearly state what they are talking about and how it fits into the previous work in the psychology field. 

Minor

4. Line 97, "that" repeated twice

Reviewer #3: In this interesting study, Gmaz and Van der Meer ask whether nucleus accumbens (NAc) neurons are responsive to context cues that establish the values predicted by subsequent "target" cues, and whether NAc neurons code for the value indicated by the combination of a prior context and current target cue. They used a simple behavior task in which the appropriate response to a target cue (go or no go) depended on the prior context cue; a delay was interposed between context cue and target cue presentation. They record from a large population of neurons in four mice, and a very strong feature of the study is that all results are reported using data from individual mice, with differences in behavior across mice described and related by the authors to differences in neural coding. In the simplest analyses, the authors find that the context cue delay period firing of a subpopulation of 5 to 13% of neurons significantly predicted the behavioral response to the target cue, and that pseudopopulation activity during this period also predicted the behavioral response above chance. The authors then apply more sophisticated population coding analyses to test whether information coding consistent with three hypothesized functions could be detected in the pseudopopulations. These functions are context coding (firing is different for different context cues until target cue presentation), state value encoding (coding of temporally-discounted reward value, which should ramp upwards from context cue presentation regardless of cue and then differ markedly as soon as a go (rewarded) vs no go (non-rewarded) target cue is presented), and what the authors call a "general cue response", which is essentially a firing response to any cue without differentiation according to meaning. They find that the "general cue response" is most strongly coded, but they also find evidence for context and state value coding. Finally, they find that context coding (during the context cue/delay period) predicts subsequent value coding (during/after target cue presentation), consistent with a gating mechanism in which the context activity modifies the subsequent target activity so that it appropriately predicts value (and/or selects the appropriate behavioral response).

This study was carefully executed and thoughtfully analyzed, and it addresses an important issue. The findings of context cue coding by NAc neurons are novel and exciting. There are however a few issues that should be resolved.

1. The major stumbling block I have to fully accepting the dPCA analysis is that the authors found many other components than the three they focus on (corresponding to general cue, state value and context coding), and these other components are only cursorily described. Two of the chosen components (state value and context) account for roughly 5% and 1-2% of the variance, and presumably the other components that are not shown each account for a similar degree of variance - or perhaps more. The impression is that the authors chose the components that happened to fit their hypothesis while ignoring the others. Perhaps some form of bootstrapping analysis would convince the reader that the components they identify with real data do not explain a similar degree of variance in randomized data. In addition, the authors should show ALL of their components in a figure similar to fig 7-supplement 1, but sorted in order from first to last, with the degree of variance explained marked, and with the author's components of interest (general, state value, context) highlighted. This would allow readers to draw their own conclusions about the relative strength of coding for the components that match the authors' hypothesis.

2. There was no imposed delay between target cue presentation and the animal's "go" response. As such, any neural response after target cue presentation could be related to movement (or witholding movement) rather than to value. The authors seem to acknowledge this, but nevertheless use "value" terminology (particularly in their description of the post-target components as "value" components). In the Discussion they should mention the absence of a delay explicitly and describe how an experiment including a delay would allow value coding to be dissociated from movement coding.

3. The authors say their findings support the hypothesized "switchboard" function of the NAc (line 354). The simplest way this function could work is if a structure such as the prefrontal cortex elevated the activity of a subpopulation of NAc neurons in one context (and perhaps depressed it in other contexts), allowing certain cues to further activate the excited neurons to produce behavior while other cues are rendered incapable of exciting the population in that context. The authors claim that their "context" coding as in the form illustrated in fig. 4A is more or less consistent with this idea, but one would expect a different response to the two target cues depending on the context - something that is not apparent either in fig. 4A and appears to be quite messy in the "real" data in fig. 7c. This raises a broader issue: if context coding subsequently influences target cue coding by some mechanism that is not a simple change in firing rate or excitation, then what is that mechanism? Given how complex such a mechanism is likely to be, how much credence should we give the idea that even though there is a correlation between context and value encoding, the two are causally related? The Discussion is not clear on this point and could be expanded to at least speculate on mechanism -- and to emphasize that without any plausible mechanism hypothesized (much less established) for this function, the observation of a mere correlation should be interpreted with caution. 

4. Fig. 3G shows a high number of "pre-context" coding units, but these units are not described in the Results and the observation is not interpreted. What exactly do these neurons significantly code for?

5. Line 167: "We found that 5-13% of units across time points were able to predict subsequent response to a target cue above chance" - did these units overlap to a large degree with the "context" coding units from Fig. 3? And what explains any degree of non-overlap?

6. It would be useful if the figures showing time on the X axis had labels for the different events (context cue on/off, etc). The convention of red and black dashed lines eventually becomes clear but additional labels on the figures themselves would be much easier on the reader.

7. Line 310 "yoked" has a specific meaning in animal behavioral science, which the authors don't intend here.

8. Typos in lines 72, 97. Line 317 fewer days, not less.

---

## [Decision Letter · Decision Letter 2]

10 Feb 2022

Dear Dr van der Meer,

Thank you for submitting a revised version of your manuscript "Context coding in the mouse nucleus accumbens modulates motivationally relevant information" for consideration as a Research Article at PLOS Biology. This revised version of your manuscript has been evaluated by the PLOS Biology editors and by the original Academic Editor and reviewers 1 and 3.

In light of the reviews (below), we are pleased to offer you the opportunity to address the remaining important points from reviewer 3 in a revised version that we anticipate should not take you very long. We will then assess your revised manuscript and your response to the reviewers' comments and we may consult the reviewers again.

We expect to receive your revised manuscript within 1 month.

Please also address the following editorial requests:

1) Abstract: Please try to make your abstract more accesible to a general life science readership. You could ask a scientist in an unrelated field to read it. 

2) Ethics: 

2.1) Please include the ID number of your protocol(s) approved by the Dartmouth College Institutional Animal Care and Use Committee (IACUC).

2.2) Please include the specific national or international regulations/guidelines to which your animal care and use protocol adhered. Please note that institutional or accreditation organization guidelines (such as AAALAC) do not meet this requirement.

3) Data:

Note that we do not require all raw data. Rather, we ask for all individual quantitative observations that underlie the data summarized in the figures and results of your paper. For an example see here: http://www.plosbiology.org/article/info%3Adoi%2F10.1371%2Fjournal.pbio.1001908#s5

3.1) We note that you mention that “All preprocessed data files are available on DataLad (http://datasets.datalad.org/, BiconditionalOdor data set)”. However, we could not find the dataset. Could you please provide a more detailed explanation? Could you also include a README file that explains how the preprocessed data were analyzed to generate all quantitative plots and graphs in all your figures, including supporting figures?

3.2) Please also ensure that each figure legend in your manuscript includes information on where the underlying data can be found.

3.3) Please ensure that your Data Statement in the submission system accurately describes where your data can be found.

**IMPORTANT - SUBMITTING YOUR REVISION**

*Resubmission Checklist*

*Published Peer Review*

*PLOS Data Policy*

*Blot and Gel Data Policy*

Sincerely,

Gabriel

Gabriel Gasque

Senior Editor

PLOS Biology

ggasque@plos.org

REVIEWS:

Reviewer #1, Kauê Machado Costa: The authors have deftly addressed all my concerns. It's a very nice study and I have no more to add at this point.

Reviewer #3: The authors have addressed most of my concerns. However, the new bootstrapping analysis used in figures 7 and S3 leaves me confused. It's very hard to find the shaded areas of the plots that indicate the range of the results using shuffled data. Presumably that's because the range was quite narrow in some cases and overlapped with the results from unshuffled data. But if that's the case, the bootstrapping analysis shows that the results with unshuffled data don't differ much from the results from shuffled data despite the authors' conclusions to the contrary. I suspect that I am simply missing something and that a more thorough explanation of the bootstrapping results (or perhaps a better visual representation of the results from shuffled vs unshuffled analyses) would clarify it.

The new figure S3 is a very useful addition. It does, however, raise a new question: there seem to be multiple components that match the expected patterns for some of the encoding types. For example, the authors chose component #3 of mouse M142 as the state value component, but to my eye component #4 looks even stronger (at least in terms of a ramping signal). The same applies to components 2 vs 3 for M146. It seems there was no objective criterion for choosing the component that *best* matched a particular pattern. In these two cases the authors chose the ones that explained the greater amount of variance, potentially biasing their conclusions.

---

## [Decision Letter · Decision Letter 3]

4 Apr 2022

Dear Dr van der Meer,

On behalf of my colleagues and the Academic Editor, Eric Nestler, I am pleased to say that we can in principle accept your Research Article "Context coding in the mouse nucleus accumbens modulates motivationally relevant information" for publication in PLOS Biology, provided you address any remaining formatting and reporting issues. These will be detailed in an email that will follow this letter and that you will usually receive within 2-3 business days, during which time no action is required from you. Please note that we will not be able to formally accept your manuscript and schedule it for publication until you have completed any requested changes.

PRESS

Sincerely, 

Richard

Richard Hodge, PhD

Associate Editor, PLOS Biology

rhodge@plos.org

PLOS
